



# Enhanced behaviors of optical properties and the radiative effects of molecular-specific brown carbon from dung combustion in the Tibetan Plateau

Qian Zhang[1,2], Yujie Zhang[1], Zhichun Wu[1], Bin Zhang[2], Yaling Zeng[3], Jian Sun[2], Hongmei Xu[2], Qiyuan Wang[4], Zhihua Li[1], Junji Cao[4], Zhenxing Shen[2*]

[1]Key Laboratory of Northwest Resource, Environment and Ecology, MOE, Xi'an University of Architecture and Technology, Xi'an 710055, China
[2]Department of Environmental Science and Engineering, Xi'an Jiaotong University, Xi'an 710049, China
[3]School of Environmental Science and Engineering, Southern University of Science and Technology, Shenzhen, 518055, China
[4]Key Lab of Aerosol Chemistry & Physics, SKLLQG, Institute of Earth Environment, Chinese Academy of Sciences, Xi'an 710061, China

Correspondence: Zhenxing Shen (zxshen@mail.xjtu.edu.cn).

**Abstract.** Traditional animal dung fuel use is a prominent source of brown carbon (BrC) in the Tibetan Plateau (TPL) region. Changes in burning conditions, fuel types, and uses of dung fuels in Plateau areas can lead to considerable uncertainties about molecular absorption properties and their radiative forcing influence on BrC. Here, the constituents of BrC's chromophoric molecules emitted from residential heating and cooking scenarios using dung fuels were proposed using ultra-high performance liquid chromatography-quadrupole time-of-flight mass spectrometry in the TPL region, China. Our results show that dung fuels in this study can release abundant BrC emissions with substantial high BrC absorption when compared with those observed in bitumite. Particularly, the linkage between BrC molecules and their absorption properties was quantified by partial least squares regression. Among detected BrC's molecular groups, above 70% of N-containing compounds (CHON and CHONS) with low-oxygen-containing and unsaturated aromatic bonds and 20.9-27.5% of CHO compounds measured from the dung combustion samples were lies in the potential BrC chromophores regions. Further, a significantly enhanced contribution of molecular absorption coefficient ($Mb_{abs}$) to total $Mb_{abs}$ (up to 99.7%) was observed in the presence of both CHO and CHON ($N_2+N_4$) with distinctive characteristics of long carbon chains and high levels of unsaturation. Interestingly, the identified CHONS markers were highly oxygenated with abundant unsaturated double-bonds and high $Mb_{abs}$, but were rarely produced under insufficient oxygen conditions at the high-altitude plateau. Meanwhile, the incomplete combustion of dung produced high values of integrated simple forcing efficiency and thus destroyed the radiation balance over the TPL region.



## 1. Introduction

The Qinghai Tibet Plateau (TPL) in China, often termed "the world's third pole", contain the largest reserve of freshwater outside the Arctic and Antarctic regions with an average altitude of 4,500m (Sills et al., 2018). In recent decades,
frequently household fuels combustion (i.e., biomass and coal) yield high atmospheric pollutants levels over TPL region, which can certainly produce an extreme sensitivity to air temperature increased and thus brings considerable harm on glacier melting, ecosystem, and even climate change on global scale (Wang et al., 2019a; Zhang et al., 2019; Chen et al., 2015; Fleming et al., 2018; Yao et al., 2012). In 2019, Zhang et al. (2022a) confirmed that the consumption of animal dung includes yak and sheep was as high as 10,990 Gg during the heating season over TPL region, which can be treated
as a major biomass resource. It's well-known that the burnings of household animal dung fuels, commonly lead to smoldering processes with low temperature and oxygen deficient conditions, are favor to produce an abundance of light-absorbing brown carbon (BrC) (An et al., 2019; Xu et al., 2022; Zhang et al., 2022a). Atmospheric BrC, as a unique of organic compounds, absorbs light significantly across the entire ultraviolet−visible (UV−Vis) wavelengths, which also brings considerable changes in atmospheric oxidation processes, atmospheric pollution and radiative forcing over TPL
region (Al-Abadleh et al., 2021; You et al., 2020; Zhang et al., 2017b; Lee et al., 2014). Thus, an understanding of the physicochemical characteristics, sources and environmental effects of BrC, a newly recognized light-absorbing organic aerosol, is fairly limited, and relevant research is urgently needed over the TPL regions.

Almost all BrC compounds contain abundances of chromophores that are often equipped with aromatic and phenolic condensed groups, unsaturated and conjugated chemical bonds (Xie et al., 2019b; Tang et al., 2020). For example, recent
field and laboratory studies have identified the compounds of nitrophenol and its derivatives (including isomers), polycyclic aromatic hydrocarbons (PAHs), oxy-PAHs, nitro-PAHs, unsaturated heterocyclic nitro products or organonitrates, benzaldehyde/benzoic acid and polycarboxylic acids as key BrC chromophores (Wang et al., 2019b; Jiang et al., 2019; Li et al., 2020a; Samburova et al., 2016; Sun et al., 2021b; Zhang et al., 2022c). However, these detected BrC chromophores includes aromatic/phenolic chemical groups only represent a few species BrC compounds
(Yuan et al., 2020; Salvador et al., 2021; Zhang et al., 2022c). Meanwhile, to date, researchers began to thoroughly investigate all possible BrC molecules using high-performance and resolution mass spectrometry instruments, however, the studies on the linkage between BrC absorption and their molecular compositions are only focus on the qualitative analysis, i.e., liner correlations (Zhou et al., 2021; Ni et al., 2021; Tang et al., 2020). Consequently, all of these newly studies would certainly limit the accurately evaluation of BrC's light-absorbing abilities and their radiative effects due
to the lack of knowledge on quantifying individual BrC's molecular absorption.

In addition, a substantial portion of BrC is derived from the sources like smoldering coal and biomass use for residential cooking or heating, fossil fuel combustions for industrial production and transportation and open burning of grass and





forest (Zhao et al., 2021; Song et al., 2019; Cheng et al., 2021; Satish and Rastogi, 2019; Zhang et al., 2021b). Meanwhile, changes in fuel type and burning conditions remain a significant influence on BrC' molecular composition which thus impacts their absorption and atmospheric chemical evolution. Wang et al. (2021a) revealed that the initial ring-breaking oxidation of aromatic species (that is, primary BrC) from incomplete burning of coal in residential areas, is a fast path to producing moderately oxidized BrC with enhanced light absorption. Interestingly, Xie et al. (2019a) found that the combination of fuel types, high relative humidity and low elemental carbon to organic carbon ratio (a measurement proxy for burn conditions) which accompany smoldering combustion, can strongly enhance the absorption ability of biomass burning BrC. On the other hand, the formation of nitrated aromatic carbon identified as strong BrC chromophores, seems to be influenced strongly by burn conditions. Therefore, to solve the substantial ambiguities in BrC absorption characteristics and their environmental impacts, the molecular structure and physical properties of the BrC chromophores from primary sources should be investigated.

In the study presented here, we analyzed comprehensively BrC's molecular absorption properties emitted from residential dung burning scenarios and compared them thoroughly with those from bitumite combustion, based on detailed molecular constituents in northeastern TPL. Therefore, we aim 1) to discuss the light-absorbing abilities of BrC; 2) to thoroughly investigate the molecular constituents, simulate molecular absorbing properties and identify specific molecular markers of BrC; and 3) to evaluate the influence of BrC absorption on the radiative effects of different residential combustion scenarios.

## 2. Experiment and Materials

### 2.1 Experimental set up.

Three typical solid fuels (yak dung, sheep dung and bitumite) of the TPL were collected and burned to inquire into the optical and chemical properties of BrC components, and they were subjected to elemental analyses (as the basis received) (Figure S1 and Table S1). Yak and sheep dung were air-dried, and bitumite was bought from local markets and kept in a closed environment. The field experiments were conducted in six typical pastoral dwellings in Gangcha County in the northeast of the TPL, where yak and Tibetan antelope are the dominant livestock types. After obtaining permission from the local herdsmen, we used iron furnaces in herdsmen's homes to distinguish the two combustion types of cooking and heating. The iron furnace is the type of stove commonly used by residents. Samples were collected with a customized dilution sampling system, the details being available in **Appendix I**. Further, the organic carbon and elemental carbon (EC) (in µg·cm$^{-2}$) of the burned samples were determined using a thermal and optical carbon analyzer (Sunset Laboratory, Portland, OR) based on a quartz filter punch and described in **Appendix II 2.1**.



## 2.2 Measurements of BrC Absorption Parameters.

As described in our previous study (Zhang et al., 2022b), we obtained the BrC fractions in methanol-soluble organic carbon (MSOC) using the solvent extraction method. First, the filter membrane samples were cut into small pieces and dissolved in a methanol solvent. Next, the filter samples were ultrasonically extracted twice for 30 minutes each time. All the extracts were then filtered through a 0.22 μm polytetrafluoron (PTFE) disposable syringe filter (Jinteng, Tianjin, China). The filtered extracts were kept away from light and stored at a low temperature for subsequent photochemical analysis. The UV-Vis absorption spectra of BrC solvents were recorded using a UV-Vis liquid waveguide capillary flow cell (LWCC, Ocean Optics, USA) at wavelengths ranging from 200 to 700 nm. In order to characterize the optical properties of BrC components, the optical absorption coefficient ($b_{abs365}$) and mass absorption efficiency (MAE$_{365}$) of BrC components at 365nm, as well as the absorption Angstrom exponent (AAE) were calculated and described in detail in **Appendix II 2.2**.

## 2.3 Detection of BrC molecules.

The composition of the combustion samples was analyzed using an ultra-high performance liquid chromatography quadrupole time-of-flight mass spectrometer (UHPLC-Q-ToF MS/MS, Waters, Milford, MA) equipped with an electrospray ionization source, operated in negative ion full-scan mode (50-2000 mass-to-charge ratio (m/z)). Before entering the instrument of UHPLC-Q-ToF MS/MS, the samples were extracted by ultrasonic methanol, filtered by a 0.22 μm PTFE injection filter, centrifuged by a cryogenic high-speed centrifuge, and then separated by a Waters Acquity UPLC HSS T3 column (1.0×100 mm, 1.8 μm, Waters Corp., Milford, MA). During the processes of BrC molecules' determination, the binary mobile phases were A (0.1% formic acid aqueous solution (v/v) and B (acetonitrile), the flow rates were 0.15 mL min$^{-1}$, the column temperature was set at 35℃, and the automatic injection volume was 3 μL. The gradient elution procedure was shown in Table S2 and the total test time for each sample was 16 min. Finally, the molecular weight of the measured material and the self-built library were determined by Unify software. It is worth noting that some screening rules were introduced to eliminate compounds that are less observable in nature (Mao et al., 2022): 0.3≤H/C≤2.5, 0≤O/C≤1.2, 0≤N/C≤0.5, and 0≤S/C≤0.2 in the ESI$^-$ mode (Lin et al., 2012a; Wang et al., 2021b). In addition, based on the BrC constituents, the calculation of emission factors of absorption (EF$_{Abs}$), double-bond equivalence (DBE), aromatic index (AI$_{mod}$), and aromatic equivalent (Xc) were used to further characterize and classify the results of the analyses (see **Appendix II 2.3).** To quantify the absorption properties of individual BrC molecules, the partial least squares regression (PLSR) model was used to process hundreds of molecules in 16 samples (Mehmood et al., 2019; Rambo et al., 2016; Zeng et al., 2020). The 5-fold cross-validation method, including 80% of training (that is calibration) samples and the external validation set with the remaining 20% of samples, was chosen to select the components of 15 for optimal fitting PLSR model performance. For each model, a series of statistical parameters were



calculated in Table S3, such as the coefficient of determination ($R^2_{cal}$ and $R^2_{val}$), the root mean square error of calibration (RMSEC), and the root mean square error of validation (RMSEV). In addition, for the predicted organic compounds in

Fig. S3, the 15 components were able to obtain optimal fitting results with an uncertainty of 1.96 Mm$^{-1}$ and a coefficient of determination ($R^2$) of 0.999. Based on the validation results, the sum of predicted BrC molecule's $b_{abs}$ values showed a good relationship with the measured BrC total $b_{abs}$ (slope = 0.51, intercept=-345.5, $R^2$ = 0.91, p < 0.001) in Fig. S4. It illustrates that the model can better interpret the majority of predicted $b_{abs}$ points at full spectrum. It is noted that a nonzero intercept in linear correlations indicates a contribution of undetected non-polar or weak polar organic

compounds.

**2.4 Simple forcing efficiency calculation.**

Bond and Bergstrom (2006) originally proposed SFE (simple forcing efficiency, W·g$^{-1}$) to estimate the radiative forcing of biomass combustion, representing the energy per unit of mass aerosol into the Earth's atmosphere (Li et al., 2020b; Wen et al., 2021). The method based on "SFE" can effectively assess the potential direct radiative forcing of BrC light

absorption emitted from solid fuel combustion in this study. However, the radiation effects caused by aerosol scattering were ignored in the estimation and only the absorption was considered (Deng et al., 2022). Therefore, the formula for calculating radiative forcing in the range of 280-700 nm can be simplified as follows (Lei et al., 2018b; Zhang et al., 2020):

$$\frac{dSFE}{d\lambda} = a_s \times \tau_{atm}(\lambda) \times (1 - F_c) \times MAE(\lambda) \times \frac{dS(\lambda)}{d\lambda}$$

where dS($\lambda$)/d$\lambda$ is the solar irradiance (W·m$^{-2}$·nm$^{-1}$), $\tau_{atm}$ ($\lambda$) is the atmospheric transmission (0.79), $F_c$ is the cloud fraction (0.6), $a_s$ is surface albedo (0.19), and MAE is the mass absorption efficiency of BrC (m$^2$·g$^{-1}$) (Chen and Bond, 2010).

**3. Results and Discussion**

**3.1 Overview of light absorption of BrC**

Fig. 1 displayed the notable differences of BrC EF$_{Abs}$ at 365 nm wavelength along with the emission factors (EFs) of organic carbon between yak and sheep combustion for cooking and heating stoves, which were also compared with those for coal combustion. The BrC EF$_{Abs}$ values reported in heating combustion ranged from 19.7 to 251.2 m$^2$·kg$^{-1}$ and averaged 130.3 m$^2$·kg$^{-1}$, whereas, relatively lower levels of BrC EF$_{Abs}$ (average: 81.4 m$^2$·kg$^{-1}$) were measured in cooking combustion scenarios (Fig. 1a). Similar to BrC EF$_{Abs}$ patterns, the OC EFs detected in combusted sheep dung, yak dung

and bitumite were 3.6±1.3, 1.9±0.4, and 2.1±0.5 g·kg$^{-1}$, respectively, which were 2.5-3.9 times higher than those for





cooking emissions (Fig. 1b). These results also indicated that same fuels yield abundant OC emissions or stronger light-absorbing BrC molecules in the heating scenario under the poor oxygen conditions than those in the cooking scenario (Sun et al., 2021a; Shen et al., 2013). Furthermore, BrC $EF_{Abs}$ varies greatly from dung to bitumite fuels even under the same conditions. The BrC $EF_{Abs}$ for the bitumite was $17.9 \pm 5.0$ m$^2 \cdot$kg$^{-1}$ for cooking and $35.2 \pm 11.5$ m$^2 \cdot$kg$^{-1}$ for heating,

which were substantially lower than for Tibetan yak dung (cooking: $75.5 \pm 13$ m$^2 \cdot$kg$^{-1}$, heating: $126.0 \pm 44.1$ m$^2 \cdot$kg$^{-1}$; p <0.01) and for Tibetan sheep dung (cooking: $182.4 \pm 69.8$ m$^2 \cdot$kg$^{-1}$, heating: $203.8 \pm 55.5$ m$^2 \cdot$kg$^{-1}$; p <0.01), indicating that the burning of dung fuels emits much more light-absorbing BrC emissions. In addition, such high BrC $EF_{Abs}$ values for household dung fuels have never been reported in previous literature, even though they were nearly 1-2 orders higher in magnitude than those reported for the residential burnings of woody fuels and crop residues in brick stoves ($3.75 \pm 3.45$

m$^2 \cdot$kg$^{-1}$) (Zhang et al., 2022b), of chunk coal, coal-clay mix and honeycomb briquettes in brick, iron stoves and Kang ($20.2 \pm 19.9$ m$^2 \cdot$kg$^{-1}$) (Zhang et al., 2021a) and of bitumite and anthracite coals with the improved and traditional stoves ($2.8$-$23.9$ m$^2 \cdot$kg$^{-1}$) (Zhang et al., 2022b). These comparisons suggested that the poor oxygen supply, high moisture, and stacking conditions in combustion scenarios increased incomplete dung combustion, thus releasing substantial amounts of strong light absorbing BrC into TPL atmospheric aerosols (Huo et al., 2018).

Fig. 2 mapped all specified samples into four optical-based BrC classes in AAE versus $\log_{10}MAE_{405}$ space. All of the dung samples in this study and the ambient BrC samples over the regions of Himalayas were skewed more toward moderately absorbing BrC (Kirillova et al., 2016). Meanwhile, the ambient BrC samples over the regions of Erenhot, northern China (Wen et al., 2021), Patiala (Srinivas et al., 2016) and New Delhi, India (Kirillova et al., 2014) lie in the mixed regions of moderately and weakly absorbing BrC, which also confirmed the dung combustion influences. These

comparisons further verified that the BrC emitted from dung combustion over TPL regions exhibited strong absorption abilities, which can be mainly attributed to the low-temperature and smoldering combustion conditions (Xu et al., 2022; Zhou et al., 2021; Saleh, 2020). In comparison, the ambient BrC samples in the region of Rondonia, Brazil with active biomass burning influences and near-source BrC samples emitted from biomass burning and coal combustion with sufficient oxygen supply, mainly fall into the region of weakly absorbing BrC (Park et al., 2020; Zhao et al., 2022; Ni et

al., 2021; Zhang et al., 2022b). These phenomena confirmed that BrC emitted from different regions and sources display different absorption abilities, which probably can be ascribed to the different constituent molecules.

In Fig. 3a, the BrC AAEs within the wavelengths of 300 to 550 nm of sheep and yak dung ($4.7$–$5.5$) in this study, were similar to those of dung-fuel cooking fires in the Indo-Gangetic Plain of southern Nepal ($4.63 \pm 0.68$) (Stockwell et al., 2016), and corn straw burning in Northeast China ($4.5 \pm 0.4$) (Wang et al., 2020). However, they were slightly lower

than the emissions from burnings grass residue and wheat stubble in the northwestern United States ($7.12$-$7.93$) (Xie et al., 2017), maize straw in the heated kang and advanced stove in Guanzhong Plain ($6.2$-$9.5$) (Lei et al., 2018b), and rice



straw and pine needles in Korea (6.9-8.2) (Park et al., 2020). These differences are mainly attributable to the differences of BrC chemical composition from individual combustion scenarios. In addition, the AAE for bitumite was significantly higher than that from burning bitumite (2.0-3.2), coal chunks (2.11-3.18), and briquettes (0.96-1.73) in most popular

stove types used in northern China (Sun et al., 2017; Zhang et al., 2022b). This may be because the abundance of methanol-soluble chromophores in BrC mainly existed in the smoldering bitumite combustion under deficient oxygen conditions over the TPL regions (Cao et al., 2021). Moreover, the AAE value of solid fuel combustion emissions in this study is similar to the emissions during the large-scale biomass burning period of Yulin (5.2 ± 0.8) (Lei et al., 2018a) and Patiala (5.1 ± 1.9) (Srinivas et al., 2016), and the environmental emissions in the Himalayas (3.8-4.2) (Kirillova et

al., 2016), which are slightly lower than the environmental emissions in Tibetan Plateau (Wu et al., 2020; Zhang et al., 2017a).

Fig. 3b compared BrC MAE values from dung and coal combustion in this study and from various typical emission sources and ambient aerosols as reported in previous studies. The average BrC MAE values in the cooking scenarios were 1.31, 0.76 and 0.95 $m^2 \cdot g^{-1}$ for yak dung, sheep dung and bitumite, respectively, which decrease by a factor of ~2

BrC MAE values in the heating scenarios. These phenomena implied that BrC emitted from cooking scenarios was characterized by stronger light absorption capacities, especially for yak dung (Tang et al., 2021). Even though the primary BrC emissions detected in this study exhibited lower MAE values than the mixed primary and secondary BrC polluted urban areas (such as Xi'an (~2.0) (Zhang et al., 2020), Beijing (1.8 ± 0.2) (Cheng et al., 2011), Nanjing (1.04 ± 0.24) (Xie et al., 2020), Los Angeles (1.9 ± 0.9) (Soleimanian et al., 2020), New Delhi (1.6 ± 0.5) (Kirillova et al., 2014), and

South Korea (Gwangju: 1.5 ± 0.3; Seoul: 1) (Park et al., 2018; Kim et al., 2016)) (see Fig. 3b). However, most average BrC MAE and AAE values for ambient aerosols over TPL regions were at low levels (MAE: 0.34-0.77 $m^2 \cdot g^{-1}$; AAE 3.8-6.24) (Kirillova et al., 2016; Wu et al., 2020; Zhang et al., 2017a). They lie in the range of values reported from Tibetan dung and coal combustion, supporting the claim that the efficiency of solar energy generation over TPL regions was being negatively impacted by BrC absorption.

**3.2 Characteristics of MSOC molecular formulas**

In Table 1, the number of 2005, 1191 and 1254 individual molecular formulas of the MSOC with numerous isomers for sheep dung, yak dung, and bitumite combustion were determined, suggesting the existence of complicated chemical compositions in these combustion scenarios. It is worth noting that a similar percentage of CHO, CHON, and CHONS compounds were observed in ESI⁻ mode in three combustion fuels, in terms of both number and relatively intensity (Fig.

S5). The sum of CHON and CHO compounds accounted for 94.3-94.6% and 79.4-88.3% of the total number and intensity, respectively, and were 7-10 times higher than CHONS. Furthermore, the percentage of S-containing compounds emitted from fresh combustion scenarios in this study was much less than those measured from ambient aerosol (21.3%-40.4%)





(Zhang et al., 2020; Bianco et al., 2018; Jiang et al., 2016). According to the measured m/z and their responded ion intensity, molecular formulas $C_cH_hO_oN_nS_s$ of MSOC were similar between heating and cooking scenarios. However,

these formulae changed slightly with different fuels, which were assigned as follows: $C_{18.3-18.8}H_{26.3-27.5}O_{1.07-1.09}N_{3.42-3.51}S_{0.14}$, $C_{18.2-18.8}H_{25.7-27.3}O_{1.06-1.07}N_{3.38-3.50}S_{0.14}$, and $C_{17.1-17.7}H_{24.6-25.1}O_{1.16-1.17}N_{3.40-3.43}S_{0.19}$ for yak dung, sheep dung, and bitumite, respectively. High C and H contents can be observed in burning dung, while the high O and S elements were slightly enriched in bitumite combustion. These results here indicates that fuel types and atmospheric conditions may be the main causes of the difference in molecular formulas while the effect of stove types can be ignored.

The differences between the DBE, $AI_{mod}$, Xc and elemental ratios of the total identified molecular formulae on the aromaticity index among each sample group were compared. In Fig.4, these detected compounds exhibited up to 40 carbon atoms and covered a wide DBE range from 0 to 20 in each combustion scenario to illustrate the distribution of BrC potential chromophores (Xu et al., 2020; Lin et al., 2018). It is clear that the majority of CHON and CHONS compounds (beyond 70%) lies in the potential BrC chromophores ranges. As supported, these N-containing compounds

with high DBE values (CHON: 7.95-8.28; CHONS: 7.32-9.85) and high $AI_{mod}$ (0.32-0.49), low O/C (<0.5) and low H/C (<1.5) ratios were referred to as low-oxygen-containing aromatic hydrocarbons with high unsaturated and aromaticity (Kourtchev et al., 2016; Wang et al., 2018). These high N-containing compounds with abundance of unsaturated bonds were probably formed from the reactions of nitrogen dioxide ($NO_2$), sulfur dioxide ($SO_2$) and nitrate radicals ($NO_3$) with phenolic compounds in dung and coal burning plumes (Xu et al., 2020). In contrast, the CHO compounds with low DBE

values of 5.42-6.04, low $AI_{mod}$ (0.16-0.21) and high H/C ratio ($\geq$1.5) were mainly concentrated in the ranges of $C_xH_{x+y}$ species, which suggested that the CHO species in the fuel combustion sample over the TPL region contain more aliphatic compounds than those from the N-containing compounds.

Based on the $AI_{mod}$ values, the molecules can be classified into five categories which displayed similar distribution among these six combustion scenarios in Fig.5. The molecules with unsaturated and low oxygen content were the most

numerous compounds (31.7-34.3%). Aliphatic compounds were the second most numerous compounds (20.0-28.3%), whereas the compounds with condensed aromatic, peptide-like and polyphenolic structures were the least numerous species, lying in the 10.5%-18.2% range. However, it is noted that the aromaticity for detected molecular groups varies greatly among specific fuel combustions scenarios. Higher content of aliphatic structures was observed in dung combustion while the molecules from bituminous coal combustion exhibited a slightly higher condensed aromatics

fraction. To clearly classify the aromatic compounds for individual molecular groups in our combustion scenarios, the Xc were calculated. For yak and sheep dung in cooking combustion, the molecules in the CHO subgroup have high Xc values (2.71 $\leq$ Xc < 2.80) which suggests they are associated with a naphthalene core structure. On the other hand, the CHON and CHONS with slightly low Xc between 2.50 to 2.59 represented an abundance of aromatic compounds with





a benzene core structure (Wang et al., 2017). Interestingly, for bituminous coal combustion, the Xc of CHONS recorded
up to 2.95 which is closely related to the pyrene core structure compounds. The Xc values indicate that most CHO and
CHON compounds emitted from the three fuels were condensed aromatics compounds with naphthalene core structures
$(2.71 \leq Xc < 2.80)$ and aromatics compounds with benzene core structures $(2.50 \leq Xc < 2.71)$. However, the CHONS
compounds from the burning of yak and sheep dung mainly contained aliphatic $(Xc < 2.50)$ compounds, while the
compounds from bituminous coal displayed distinct aromaticity in relation to high Xc (2.59), DBE (8.85) and $AI_{mod}$
(0.46) values.

**3.3 Linkage between light absorptions and molecules**

As can be seen from Fig. 6a and b, the value and percentage of simulation $Mb_{abs}$ for the total CHON compounds were
in the range of 370.1-1815.1 $Mm^{-1}$ and 47.2-69.9%, respectively, which were higher than those for the CHO (194.8-
1191.1 $Mm^{-1}$; 24.5-41.9%) and CHONS (4.60-225.7 $Mm^{-1}$; 0.3-26.6%) compounds. The values confirm that CHON
compounds were likely to be the dominant light absorbers in solid fuel combustion over the TPL region. Furthermore,
the $Mb_{abs}$ values of individual chemical groups varies considerably across in different combustion scenarios. For example,
the sum of CHO and CHON compounds contributed up 99.7% of the total $Mb_{abs}$ in sheep and yak dung combustion. In
depth, these absorbing CHON compounds were ranged between 0.3 < H/C < 2, 0 < O/C < 0.9 and high Xc, showing
their low degrees of oxidation and saturation, especially in the sheep dung heating scenario. In contrast, the simulated
$Mb_{abs}$ values of CHONS compounds possessed low fractions, which only increased significantly to 16.5% and 26.6% in
the cooking and heating combustion of bituminous coal, respectively.

Considering all combustion experiments, the specific CHO, CHON and CHONS compounds with high relatively
intensity and simulated $Mb_{abs}$ values are identified in Table 2. All of the selected N-containing compounds (CHON and
CHONS) with the high fractions of relative intensity and $Mb_{abs}$ presented high DBE and Xc values, which confirmed
high degrees of aromaticity and unsaturation. The compounds of $C_{16}H_{18}O_3N_2$, $C_{18}H_{30}O_2N_4$, and $C_{22}H_{28}O_4N_2$ possessed
large fractions in terms of relative intensity (>3%) and $Mb_{abs}$ (>3%) for the CHON group detected in dung combustion,
while $C_{32}H_{46}O_3N_2$, $C_{20}H_{26}O_2N_2$ and $C_{16}H_{26}O_3N_2$ recorded high levels from mixed sources of dung and bitumite
combustion samples. In addition, the reduced-N compounds (O/N < 3) of $N_2$ and $N_4$ CHON, which varied particularly
in different combustion scenarios. These N-compounds were equipped with the abundance of C=N and C-N functional
group, which possibly closely related their various molecular structures and transforming paths (Zeng et al., 2021). For
example, the $C_{20}H_{26}O_2N_2$ compound, a secondary oxidation product of primary emitted compounds (e.g., phenols and
cresol) in smoke plumes, contributed 52.5-59.5% and 21.9-38.6% to the total relative intensity and $Mb_{abs}$, respectively,
in dung burning for cooking, while the compound sharply decreased to 23.8-28.0% and 6.1-7.2% in the dung heating
scenarios.



The results indicate that combustion types can determine the abundance of BrC compounds. The $C_{26}H_{24}ON_4S$ compound is likely formamide, which consists of the most light-absorbing BrC chromophores in the CHONS group and produced values up to 17.3%, 48.3% and 59.4%, respectively, in the burning samples of yak, sheep and bituminous coal. The remaining CHONS formulas with high DBE and $Mb_{abs}$ detected uniquely in samples were highly oxygenated in the $O_2S$, $O_3S$, and $O_4S$ subclasses, which possessed enough oxygen atoms to allow assignment of $-OSO_3H$ and/or $-ONO_2$ groups

in their formulas and thus can be regarded as OSs or nitrooxy-OSs (Lin et al., 2012b; O'brien and Kroll, 2019). However, the emissions of highly oxygenated CHONS would certainly be limited due to the oxygen deficient environment in the TPL region, which in turn would reduce the absorbance effect of CHONS in the combustion samples. Also, the light-absorbing CHO compounds can be identified as unsaturated aromatic compounds (UA-CHO, H/C<2 and DBE≥3). Four typical UA-CHO compounds were mainly found in dung samples, accounting for 2.8%-16.1% of the total $Mb_{abs}$ of

individual CHO groups. All of the UA-CHO compounds contain abundant conjugated compounds and are high in carbon atoms (16-31), which were assigned to the long-carbon aromatic compounds. Meanwhile, the existence of identified tracers was further examined. The compound of $C_{31}H_{40}O_3$ was likely formed from phenol (Lin et al., 2016), which can be selected as important markers for TPL residential combustion. In contrast, the compounds of $C_{18}H_{18}O_4$, $C_{19}H_{22}O_3$ and $C_{22}H_{32}O_2$ likely responded to succinic acid-4-biphenyl ethyl ester, p-methoxybenzoic acid-5-phenylpentyl ester, and 4-

ethylbenzoic acid-tridec-2-ynyl ester, respectively, with several isomers which contained one or two benzene rings and -COOH that were regarded as unique markers for dung samples (Graham et al., 2002).

### 3.4 Estimation of the potential radiative effects of the identified molecules

To fully evaluate radiative forcing relative to BrC from solid fuel combustion over the TPL region, the BrC SFE values were estimated over wavelengths of 280 to 700 nm. In Fig.S6a, the integrated average SFE values in the entire solar spectral ranges were 5.3±2.8, 4.1±1.9, and 3.5±1.5 W·g$^{-1}$ for the burning emissions of yak, sheep, and bituminous coal,

respectively, which showed a similar trend with EF$_{Abs}$ values. In addition, the SFE peaks showed bimodal distribution in the ultraviolet bands (i.e., around 400 nm) in the spectra of the samples collected in these three fuels. In Fig. S6b, our SFE$_{BrC}$ values from burning residential solid fuels over the TPL region were lower than paddy-residue burning over the Indo-Gangetic Plain (Choudhary et al., 2021) and rice straw burning in Anhui Province, China (Zhao et al., 2022), but they were comparable with those from the burning of maize straw in Shaanxi province (Lei et al., 2018b) and from

ambient aerosols over heavily polluted northern Chinese cities, including Tianjin, Erenhot, Zhangbei and Jinan (Wen et al., 2021; Deng et al., 2022). These phenomena confirmed that the heavy BrC loadings from residential fuels can cause high radiative forcing effects over TPL regions.

Even using the same fuels, the comparatively higher variability of the BrC SFE from these solid fuels was still displayed

in the cooking and heating scenarios (Fig. 6c and d). In Fig. S6 c and d, the integrated BrC SFEs at the wavelength of





300-400 nm of dung and bitumite fuels in cooking combustion, were 1.5-2 times higher than those in heating combustion. The most probable cause for the variance is that these fuels are more likely to produce instantaneous and incomplete combustion in cooking combustion than bituminous fuel during heating combustion scenarios, so BrC's light-absorbing capacities can be strengthened. For heating combustion, the integrated BrC SFEs at 300-400nm of yak and sheep dung were 1.5±0.8 and 1.4±0.8, respectively, which were 1.6 times higher than that for bitumite. Owing to the dung fuels' small specific surface area (Figure S1c and d), the high EFs of BrC absorption can be observed (Zhang et al., 2022a; Sun et al., 2021a), which mainly yielded much higher BrC SFE.

## 4 Conclusion

We all know that burning biofuels of yak and sheep dung is still the most traditional and popular way of heating and cooking over the TPL region. The total consumption of both yak and sheep dung can even possess up to nearly 70% of the total dung fuel consumptions during heating periods (Zhang et al., 2022a), resulting in severe BrC pollution. Former field studies on the TPL BrC were mostly focused on evaluating their total light absorption capacity and primarily identifying BrC's possible sources but were limited in quantifying BrC's molecular constituents and the related absorption properties from animal dung combustion. Our current study explored differences in molecular characteristics of BrC between two specific animal dung combustions in traditional stoves and compared with those from bitumite combustion. We recognized that the highest BrC $EF_{Abs}$ and SFE from incompletely burned dung fuels is respectively 4-9 and 1.5-1.6 times higher than those from bitumite fuels. Furthermore, the compounds of $N_2$ and $N_4$ CHON with high unsaturated aromaticity and CHO with benzene rings and -COOH, were unique markers for dung-fuel based BrC. In comparison, although the high oxygenated CHONS compounds presented high molecular absorption, their emissions were small as a result of the anoxic combustion environment at the high-altitude plateau. Specifically, the unique CHO markers from dung combustion can be identified as $C_{18}H_{18}O_4$ and $C_{19}H_{22}O_3$ with adipic acid and 2-methylvaleric acid, respectively. Therefore, this work provides exhaustive molecular information on the TPL's BrC emission inventory with the influence of various dung fuel and stove combinations including oxygen-deficient combustion, which can better explain the reasons behind high BrC emissions and their radiative efficiency.



*Data availability.* Requests for all data in this study and any questions regarding the data can be directed to Qian Zhang (zhangqian2018@xauat.edu.cn) or Zhenxing Shen (zxshen@xjtu.edu.cn).

*Competing interests.* The authors declare that they have no conflict of interest.

*Special issue statement.* This article is part of the special issue *"In-depth study of the atmospheric chemistry over the Tibetan Plateau: measurement, processing, and the impacts on climate and air quality (ACP/AMT inter-journal SI)"*. It
is not associated with a conference.

*Author contributions.* QZ: Writing - original draft, Visualization; YZ: Data curation and analysis; ZW: Experiment, Formal analysis; BZ: Sample collection; YZ: Data curation, Validation; JS: Data curation, Investigation; HX: Data curation, Validation; QW: Writing - review & editing; ZL: Writing - review & editing; JC: Resources, Visualization; ZS: Conceptualization, Supervision, Project administration.

*Financial support.* This research was financially supported by the National Natural Science Foundation of China (42007193, 41877383, and 21661132005), Key Research and Development Projects of Shaanxi Province, China (2022ZDLSF06-07), a grant from SKLLQG, Chinese Academy of Sciences, China (SKLLQG 2028).

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

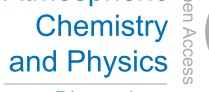

**Table 1.** Individual molecular formulas of the BrC with numerous isomers for each combustion scenario.

| Samples | Combustion | Parameters | All | CHO | CHON | CHONS |
|---|---|---|---|---|---|---|
| Yak dung | Cooking | number | 755 | 342 | 372 | 41 |
| | | molecular weight | 332.21 | 354.33 | 309.42 | 354.55 |
| | | $AI_{mod}$ | 0.30 | 0.16 | 0.42 | 0.32 |
| | | DBE | 6.87 | 5.42 | 8.11 | 7.73 |
| | | O/C | 0.20 | 0.20 | 0.20 | 0.22 |
| | | H/C | 1.42 | 1.59 | 1.29 | 1.34 |
| | | Xc | 2.60 | 2.77 | 2.51 | 2.54 |
| Sheep dung | | number | 444 | 211 | 206 | 27 |
| | | molecular weight | 321.47 | 329.70 | 307.19 | 366.10 |
| | | $AI_{mod}$ | 0.31 | 0.19 | 0.42 | 0.47 |
| | | DBE | 7.10 | 5.60 | 8.28 | 9.85 |
| | | O/C | 0.21 | 0.20 | 0.21 | 0.23 |
| | | H/C | 1.40 | 1.55 | 1.28 | 1.15 |
| | | Xc | 2.58 | 2.67 | 2.50 | 2.59 |
| Bitumite | | number | 658 | 270 | 354 | 34 |
| | | molecular weight | 319.25 | 338.59 | 305.66 | 307.17 |
| | | $AI_{mod}$ | 0.37 | 0.21 | 0.49 | 0.32 |
| | | DBE | 7.29 | 6.04 | 8.23 | 7.32 |
| | | O/C | 0.22 | 0.22 | 0.21 | 0.24 |
| | | H/C | 1.35 | 1.50 | 1.23 | 1.34 |
| | | Xc | 2.68 | 2.85 | 2.53 | 2.95 |
| Yak dung | Heating | number | 436 | 192 | 221 | 23 |
| | | molecular weight | 324.00 | 338.02 | 310.81 | 334.29 |
| | | $AI_{mod}$ | 0.31 | 0.19 | 0.41 | 0.36 |
| | | DBE | 6.95 | 5.66 | 7.95 | 8.00 |
| | | O/C | 0.21 | 0.20 | 0.21 | 0.21 |
| | | H/C | 1.41 | 1.56 | 1.30 | 1.30 |
| | | Xc | 2.51 | 2.72 | 2.45 | 2.38 |
| Sheep dung | | number | 1561 | 725 | 748 | 88 |
| | | molecular weight | 331.63 | 349.78 | 311.62 | 351.63 |
| | | $AI_{mod}$ | 0.30 | 0.17 | 0.42 | 0.36 |
| | | DBE | 6.94 | 5.49 | 8.19 | 8.18 |
| | | O/C | 0.20 | 0.21 | 0.20 | 0.21 |
| | | H/C | 1.42 | 1.57 | 1.29 | 1.30 |





| | | | | | |
|---|---|---|---|---|---|
| | Xc | 2.60 | 2.73 | 2.50 | 2.45 |
| | number | 596 | 248 | 314 | 34 |
| | molecular weight | 323.10 | 340.26 | 310.18 | 317.21 |
| Bitumite | $AI_{mod}$ | 0.36 | 0.21 | 0.47 | 0.46 |
| | DBE | 7.35 | 6.01 | 8.25 | 8.85 |
| | O/C | 0.21 | 0.21 | 0.21 | 0.23 |
| | H/C | 1.35 | 1.51 | 1.25 | 1.16 |
| | Xc | 2.62 | 2.77 | 2.52 | 2.59 |




**Table 2.** Possible structural features of the molecular formula of the high light-absorbing BrC detected by mass spectrometry.

| Type | Molecular formula | Source | Category | Possible name | Potential structure (https://webbook.nist.gov/chemistry/form-ser/) | $Mb_{abs}$ | DBE | Xc | Fraction ($Mb_{abs}$) | Fraction (Relative intensity) |
|---|---|---|---|---|---|---|---|---|---|---|
| | $C_{32}H_{46}O_3N_2$ | Yak dung | Cooking | 4-n-Pentanoyl-4-n'-pentadecanoyloxyazobenzene | | 368.24 | 11 | 2.79 | 6.91% | 3.09% |
| | | Bitumite | | | | 297.98 | 11 | 2.79 | 6.74% | 3.05% |
| | $C_{22}H_{28}O_4N_2$ | Yak dung | Cooking | O-nitro carbanilic acid | | 420.72 | 10 | 2.75 | 7.90% | 4.23% |
| | $C_{16}H_{18}O_3N_2$ | Yak dung | Cooking | Diazene, bis(4-ethoxyphenyl)-, 1-oxide | | 160.29 | 9 | 2.73 | 3.01% | 3.46% |
| CHON | $C_{20}H_{26}O_2N_2$ | Yak dung | Cooking | Hydroquinidine | | 2055.44 | 9 | 2.75 | 38.59% | 59.46% |
| | | Sheep dung | | | | 1450.33 | 9 | 2.75 | 21.90% | 52.46% |
| | | Bitumite | | | | 1491.27 | 9 | 2.75 | 33.75% | 52.59% |
| | | Yak dung | Heating | | | 183.77 | 9 | 2.75 | 6.05% | 28.01% |
| | | Sheep dung | | | | 955.46 | 9 | 2.75 | 7.20% | 23.80% |
| | | Bitumite | | | | 864.95 | 9 | 2.75 | 13.71% | 26.34% |
| | $C_{18}H_{30}O_2N_4$ | Yak dung | Cooking | / | / | 1409.34 | 6 | 2.60 | 26.46% | 3.53% |
| | | Sheep dung | | | | 558.97 | 6 | 2.60 | 8.44% | 4.75% |
| | | Yak dung | Heating | | | 1509.12 | 6 | 2.60 | 49.65% | 19.91% |
| | | Sheep dung | | | | 1409.34 | 6 | 2.60 | 10.61% | 3.04% |
| | $C_{16}H_{26}O_3N_2$ | Sheep dung | Cooking | Benzoic acid, 3-amino-4-propoxy-, 2-(diethylamino)ethyl ester | | 2932.26 | 5 | 2.43 | 44.27% | 13.15% |
| | | Bitumite | | | | 707.70 | 5 | 2.43 | 16.02% | 3.09% |
| | | Yak dung | Heating | | | 569.71 | 5 | 2.43 | 18.74% | 10.76% |
| | | Sheep dung | | | | 3172.64 | 5 | 2.43 | 23.90% | 9.80% |
| | | Bitumite | | | | 2915.80 | 5 | 2.43 | 46.21% | 11.01% |
| CHONS | $C_{26}H_{24}ON_4S$ | Yak dung | Cooking | Formamide, n-benzyl-n-[3-benzyl-6-benzylamino-4-thioxo-5-pyrimidinyl]- | | 74.65 | 17 | 2.88 | 63.78% | 17.30% |
| | | Sheep dung | | | | 236.15 | 17 | 2.88 | 79.77% | 48.33% |
| | | Bitumite | | | | 293.51 | 17 | 2.88 | 93.22% | 59.41% |
| | | Yak dung | Heating | | | 6.05 | 17 | 2.88 | 59.49% | 18.06% |
| | | Sheep dung | | | | 225.52 | 17 | 2.88 | 79.88% | 32.09% |
| | | Bitumite | | | | 321.49 | 17 | 2.88 | 46.89% | 28.58% |


| | Formula | Source | Process | Name | Structure | | | | | |
|---|---|---|---|---|---|---|---|---|---|---|
| | C₁₉H₁₈O₂N₂S | Yak dung | Cooking | P-(p-tolylsulfonylamido-)diphenylamine | | 3.95 | 12 | 2.81 | 3.38% | 7.83% |
| | | Yak dung | Heating | | | 0.39 | 12 | 2.81 | 3.86% | 10.03% |
| | C₁₂H₈O₄N₂S | Yak dung | Cooking | di(p-Nitrophenyl) sulfide | | 17.45 | 10 | 2.73 | 14.91% | 11.10% |
| | | Sheep dung | | | | 26.76 | 10 | 2.73 | 9.04% | 15.02% |
| | | Yak dung | Heating | | | 0.53 | 10 | 2.73 | 5.17% | 4.31% |
| | | Sheep dung | | | | 6.45 | 10 | 2.73 | 2.29% | 2.52% |
| | | Bitumite | | | | 58.99 | 10 | 2.73 | 8.60% | 14.39% |
| | C₁₇H₂₆O₄N₂S | Yak dung | Cooking | Sultopride | | 4.32 | 6 | 2.43 | 3.69% | 20.99% |
| | C₁₀H₁₆O₃N₂S | Bitumite | Cooking | 1H-Thieno[3,4-d]imidazole-4-pentanoic acid, hexahydro-2-oxo-, (3aS,4S,6aR)- | | 7.17 | 4 | 2.00 | 2.28% | 18.26% |
| CHO | C₃₁H₄₀O₃ | Sheep dung | Cooking | Phenol, 2,6-bis[[3-(1,1-dimethylethyl)-2-hydroxy-5-methylphenyl]methyl]-4-methyl- | | 605.23 | 12 | 2.81 | 15.30% | 22.90% |
| | | Sheep dung | Heating | | | 854.89 | 12 | 2.81 | 5.98% | 13.43% |
| | | Bitumite | | | | 791.88 | 12 | 2.81 | 13.53% | 21.01% |
| | C₁₈H₁₈O₄ | Sheep dung | Cooking | Succinic acid, 4-biphenyl ethyl ester | | 244.99 | 10 | 2.75 | 6.19% | 7.67% |
| | C₁₉H₂₂O₃ | Yak dung | cooking | p-Methoxybenzoic acid, 5-phenylpentyl ester | | 474.44 | 9 | 2.73 | 7.17% | 11.02% |
| | C₂₂H₃₂O₂ | Yak dung | Cooking | 4-Ethylbenzoic acid, tridec-2-ynyl ester | | 1098.82 | 7 | 2.67 | 16.62% | 8.59% |
| | | Sheep dung | | | | 885.79 | 7 | 2.67 | 22.39% | 8.54% |
| | | Yak dung | Heating | | | 206.31 | 7 | 2.67 | 7.49% | 8.48% |





**Figure Captions:**

**Figure 1.** Comparisons of (a) $EF_{Abs}$, (b) EFs for organic carbon ($EF_{OC}$) among six combustion scenarios.

**Figure 2.** Plot of the optical-based classification scheme representative of operationally defined BrC classes in the log10($MAE_{405nm}$) – AAE space. The shaded regions represent very weakly absorbing BrC (VW-BrC), weakly absorbing BrC (W-BrC), moderately absorbing BrC (M-BrC), and strongly absorbing BrC (S-BrC).

**Figure 3.** Comparison of MAE(a) and AAE(b) in source emissions and ambient aerosol BrC.

**Figure 4.** Plot of the double bond equivalent (DBE) vs the number of carbon atoms in identified species detected in six combustion models. Lines indicate DBE reference values of linear conjugated polyenes $C_xH_{x+2}$ (green solid line), *cata*-condensed PAHs (kelly solid line), and fullerene-like hydrocarbons with DBE = $0.9 \times c$ (orange solid line). Data points inside the blue shaded area are potential BrC chromophores. The pie chart shows the counts of CHO (red), CHON (dark blue), and CHONS (earthy yellow) in the blue shaded area.

**Figure 5.** The distribution of $AI_{mod}$ values different component emitted from three fuels used in the cooking and heating scenarios. The distribution of alkanes ($AI_{mod}=0$), highly unsaturated compounds ($0<AI_{mod}\leq0.50$, H/C<1.5), unsaturated aliphatic compounds ($0<AI_{mod}\leq0.50$, H/C ≥ 1.5), highly aromatic compounds ($0.5<AI_{mod}\leq0.67$), and condensed aromatic ($AI_{mod}>0.67$) is shown for each compound class of the MSOC in the six combustion scenarios.

**Figure 6.** The bar graph shows the simulated $Mb_{abs}$ values and their relative contributions (%) for the six combustion modes.





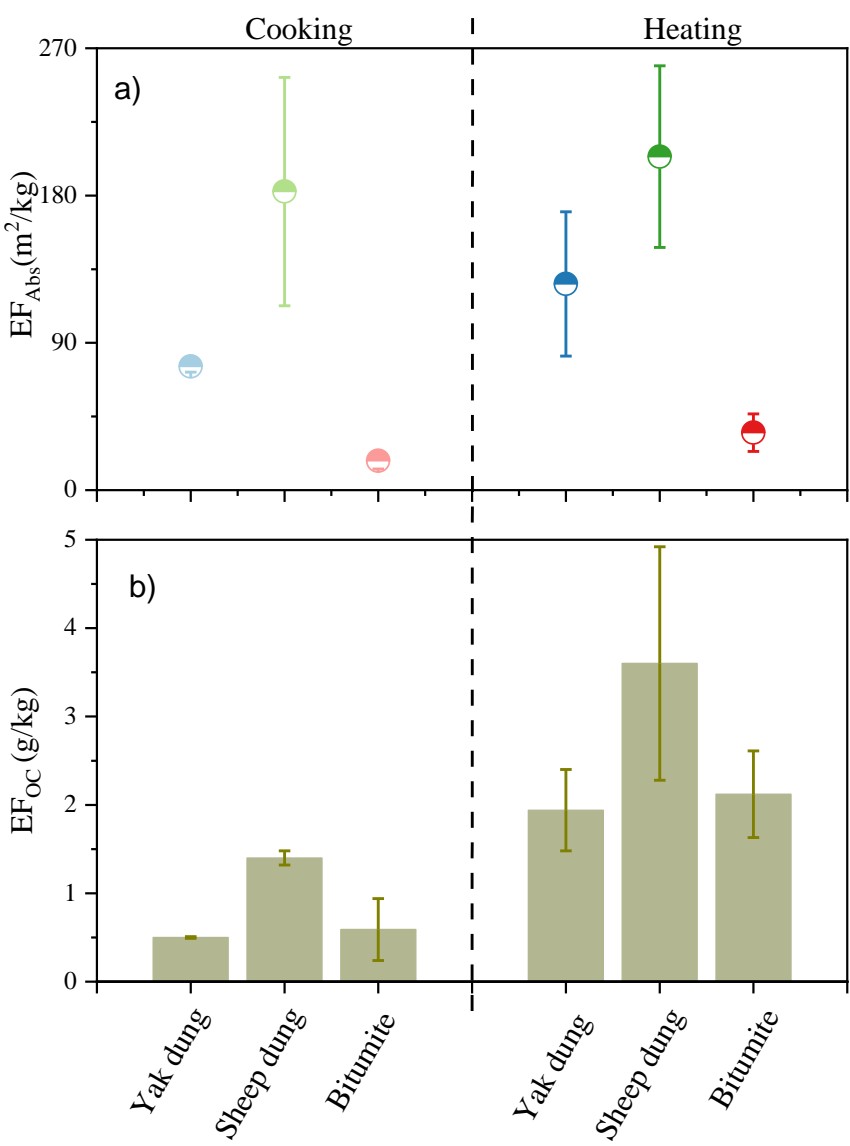

**Figure 1.** Comparisons of (a) EF$_{Abs}$, (b) EFs for organic carbon (EFOC) among six combustion scenarios.


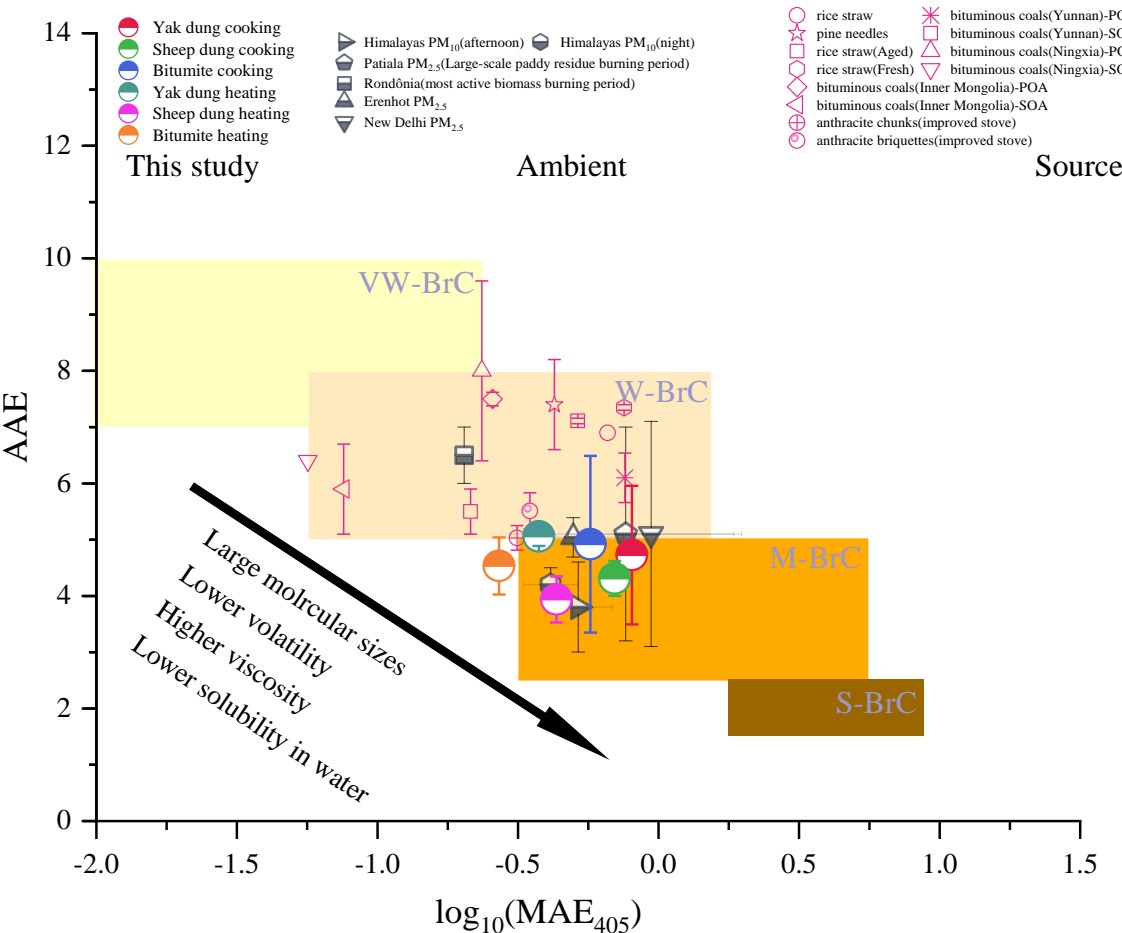

**Figure 2.** Plot of the optical-based classification scheme representative of operationally defined BrC classes in the log10(MAE$_{405\,nm}$) – AAE space. The shaded regions represent very weakly absorbing BrC (VW-BrC), weakly absorbing BrC (W-BrC), moderately absorbing BrC (M-BrC), and strongly absorbing BrC (S-BrC).



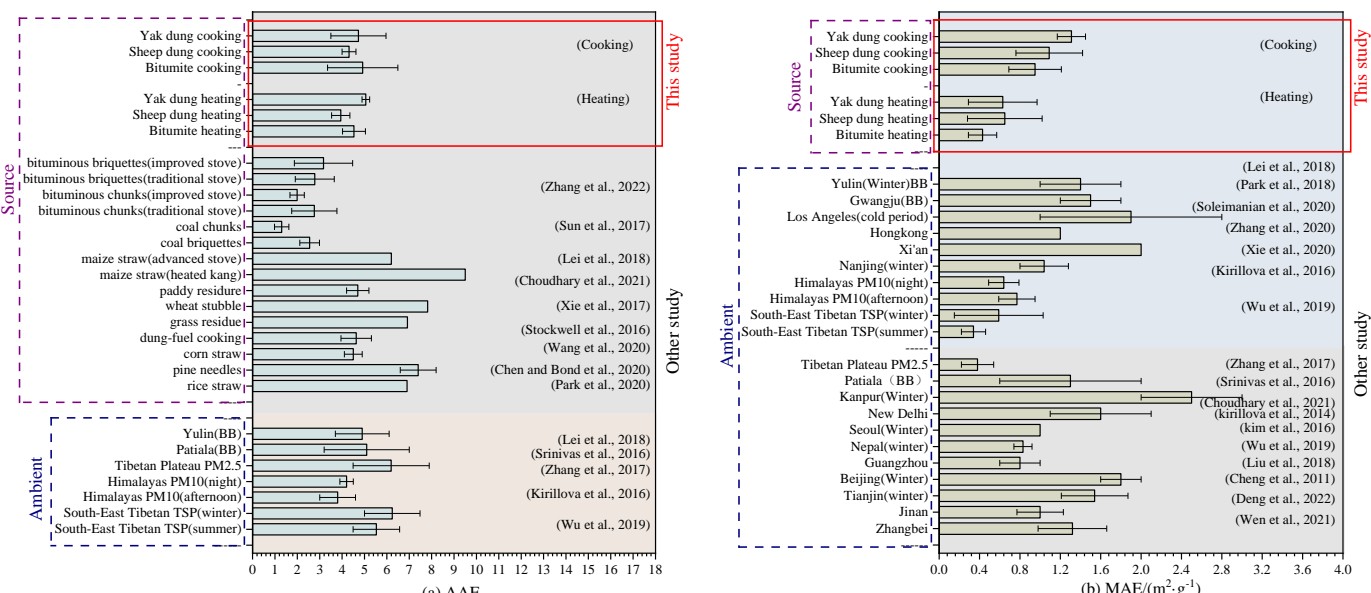

**Figure 3.** Comparison of MAE(a) and AAE(b) in source emissions and ambient aerosol BrC

**Figure 4.** Plot of the double bond equivalent (DBE) vs the number of carbon atoms in identified species detected in six combustion models. Lines indicate DBE reference values of linear conjugated polyenes $C_xH_{x+2}$ (green solid line), *cata*-condensed PAHs (kelly solid line), and fullerene-like hydrocarbons with DBE = $0.9 \times c$ (orange solid line). Data points inside the blue shaded area are potential BrC chromophores. The pie chart shows the counts of CHO (red), CHON (dark blue), and CHONS (earthy yellow) in the blue shaded area.





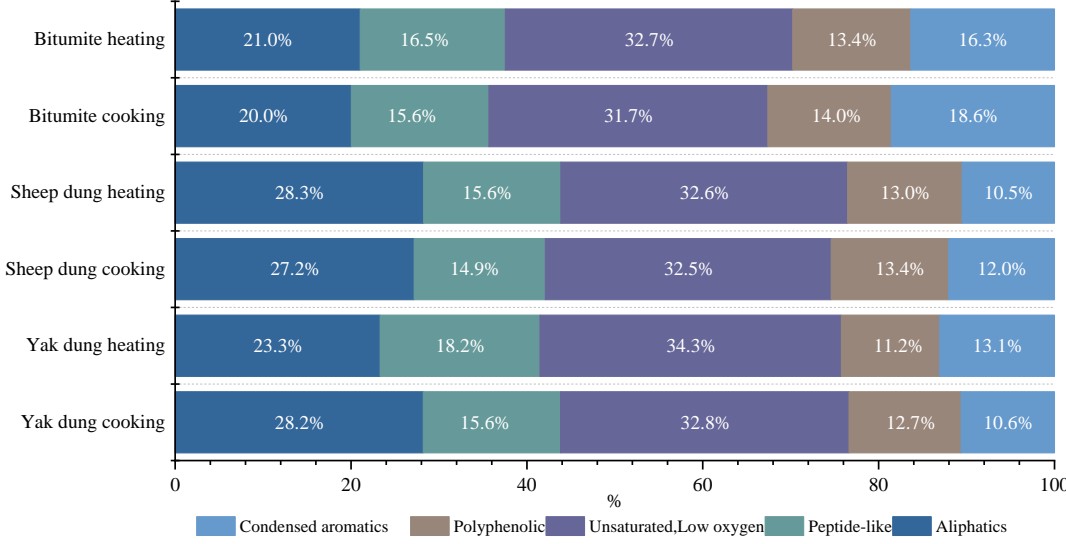

645

**Figure 5.** The distribution of $AI_{mod}$ values different component emitted from three fuels used in the cooking and heating scenarios. The distribution of alkanes ($AI_{mod}=0$), highly unsaturated compounds ($0<AI_{mod}≤0.50$, H/C<1.5), unsaturated aliphatic compounds ($0<AI_{mod}≤0.50$, H/C ≥ 1.5), highly aromatic compounds ($0.5<AI_{mod}≤0.67$), and condensed aromatic ($AI_{mod} >0.67$) is shown for each compound class of the MSOC in the six combustion scenarios.

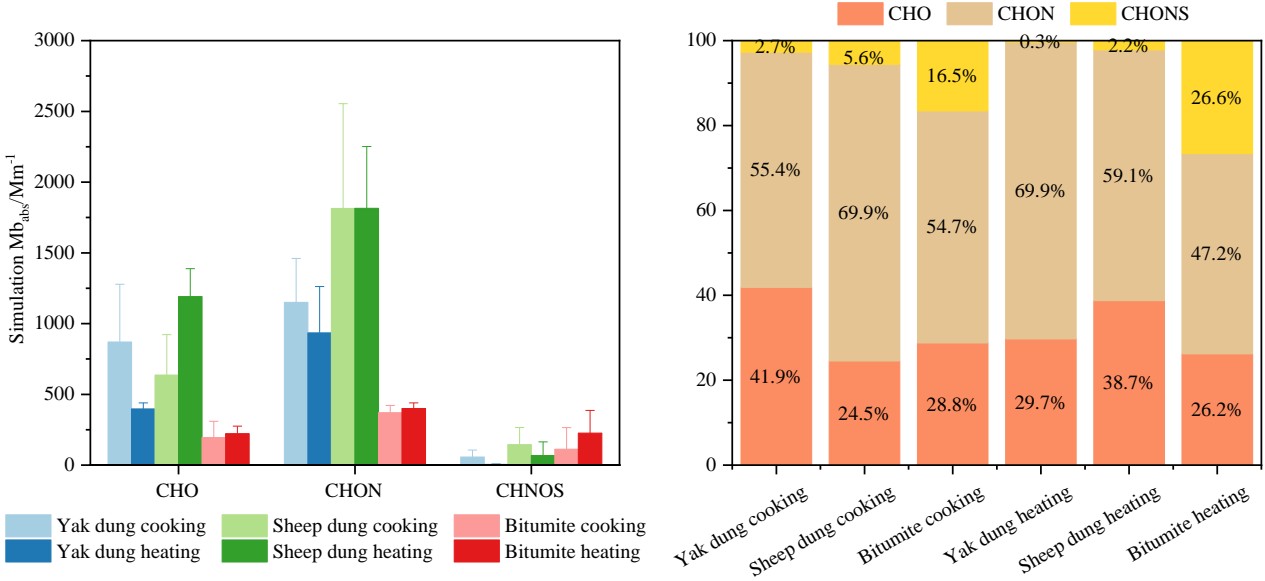

650

**Figure 6.** The bar graph shows the simulated $Mb_{abs}$ values and their relative contributions (%) for the six combustion modes.