# Peer review of "Enhanced behaviors of optical properties and the radiative effects of molecular-specific brown carbon from dung combustion in the Tibetan Plateau"

_Atmospheric Chemistry and Physics, 2022_

## Author Comment (AC1)

**RC1**: 'Comment on acp-2022-801', Anonymous Referee #1, 12 Jan 2023  reply

This manuscript, prepared by Qian Zhang et al., describes an analysis of Brown Carbon (BrC) emitted from burning traditional solid fuels (yak dung, sheep dung, and bitumite) commonly used in the Qinghai Tibet Plateau (TPL) region. With the measurement of light absorptivity and molecular-level information, the authors attempted to identify major light-absorbing species and markers for such combustion. TPL is a climatically sensitive region, and addressing light-absorbing climate forcers, such as BrC, emitted in this region is of paramount importance. Particularly, emissions arising from traditional (but still widely practiced) solid fuels, including dung burning, are significantly understudied in the literature. In this regard, the research topic is important, timely, and fits the scope of ACP. In particular, I find the observation that these emissions are particularly high in BrC emission fascinating.

 However, I cannot recommend publishing this manuscript in its current form. I have a number of major scientific concerns, as listed below, and I think parts of the discussion just seem to be incomplete in the current version. It may be potentially publishable after significant rewriting.  Last but not least, the literary grade of the manuscript is not excellent, with countless grammatical errors. I would recommend the authors undergo additional editing and proofreading.

**Response:** We thank the reviewer's comments on our manuscript. We have added detailed descriptions of the methodology used in this study including the MSOC molecules detection (see the revised sections 2.2 and 2.3), the molecular absorption coefficient ($Mb_{abs}$) of detected molecules calculation (see the revised section 2.4), the BrC molecules identification (see the revised section 2.4) and burning conditions classification (see Table S4 and section 3.1 in the revised supplementary). In addition, the revised descriptions of the detected molecular formulae, $Mb_{abs,}$ and MCE conditions influences were added to the revised manuscript. Our point-by-point responses for most of the suggestions provided by the reviewer are detailed below. Further, the language was also polished by a native English speaker. We hope these revisions will satisfy the reviewer for publishing the manuscript.

**Major comment**

My biggest concern is that the analytical method used, UHPLC-ToF-MS/MS, has limitations in both quantitation and identification of molecules. Although I acknowledge that the chemical analyses were done carefully, and with much appreciated chemical insights, the authors not only did not carefully discuss these limitations but seem to be unaware of them. Please see my breakdown below.

**Response**: Yes, the detailed interpretations on the error of MSOC extraction (see the revised section 2.2 and supplementary section 2.2), the limitations of negative ESI mode measurement (see lines 114-116 in the revised section 2.3), and the processes of MSOC molecular formulae identification (see lines 123-133 in the revised section 2.3) were added in the revised manuscript. We have made most of the changes suggested by the reviewer, both to the text and figures, in response to the breakdown below.

Completeness of the detected species

There are multiple reasons the method used here can detect only a fraction of BrC emitted. 1) methanol cannot extract all BrC compounds on the filter 2) The column cannot elute all the compounds in MSOC. 3) ESI(-) only detects compounds with an acidic proton, which is far from the entire spectrum of MSOC compounds.

**Response**: Yes, the method used in the manuscript can detect only partial BrC compounds in relation to the selection of solvent, column and ESI source.

It's noted that methanol was verified as a high-efficiency method for extracting more than 90% organic carbon (Chen et al., 2019; Cheng et al., 2016; Liu et al., 2013). Chen and Bond (2010) indicated that a larger portion of the absorption that comes from OC is extractable only by methanol. Therefore, in our previous studies, the MSOC fraction is generally used as a surrogate for BrC (Shen et al., 2017; Zeng et al., 2020). To clarify this limitation, the descriptions of uncertainties of methanol extraction were added to Section 2.2 in the revised manuscript as below.

*Lines 99-103:*

*"Methanol is an efficient solvent for extracting OC from the quartz fiber filter, but there is still 3-15% OC that could not be extracted (Chen et al., 2019; Cheng et al., 2016; Liu et al., 2013). In this study, the absorption of methanol-soluble OC (MSOC) is used as a surrogate for the characteristics of total BrC, which might lead to a certain degree of underestimation of the BrC light absorption and constituents. The MSOC fraction was obtained using the solvent extraction method as follows."*

For characterizing MSOC molecules, the ultrahigh-resolution mass spectrometer such as FT-ICR MS, Orbitrap MS, and UHPLC-Q-TOF MS/MS coupled with negative ESI source had been widely used and reported in recent years (An et al., 2019; Wang et al., 2017; Zeng et al., 2020). ESI mode preferentially ionizes semi-polar and polar molecules in OC such as nitro-aromatics, aromatic acids, and carboxylic acids (Kujawinski, 2002; Zeng et al., 2021). These detected polar OC equipped with light absorption and acid properties would affect deeply many acid-catalyzed heterogeneous reactions in the atmospheric condensed phase and change the hygroscopic properties and optical properties of aerosols and are worth to be understood. As for $ESI^+$ and $ESI^-$ modes comparison, Lin et al. (2018) showed that, during the processes of biomass burning, around 65% of the compounds detected with the $ESI^-$ mode located at the "BrC domain" region, whereas only 24% of the detected compounds was within the "BrC domain" with the $ESI^+$ mode. The results imply that $ESI^+$ mode preferentially ionizes non-BrC polar compounds. Furthermore, the HSS T3 column used in the UHPLC TOF MS/MS is known to retain and separate polar compounds that even have a wide range of polarity (Chauveau-Duriot et al., 2010). Thus, in this study, we chose the UHPLC HSS T3 column coupled with $ESI^-$ source as a priority method to well-understand the polar BrC domain compounds from dung combustion in the Tibetan Plateau.

However, as the reviewer mentioned, the rest BrC compounds including weak polar or non-polar compounds cannot be detected using ESI mode, which was not discussed in this study. Therefore, to express the detection of $ESI^-$ mode explicitly, the sentence that describes the limitation of $ESI^-$ sources has been added to in Section 2.3 in the revised manuscript as follows.

*Lines 114-116:*

*"In this study, the negative ESI UHPLC-Q-TOF MS/MS can measure semi-polar and polar organic molecules with acidic protons, but it is insensitive to non-polar compounds. Therefore, the detected molecules in our study only refer to a part of the MSOC fraction."*

References:

*An, Y., Xu, J., Feng, L., Zhang, X., Liu, Y., Kang, S., Jiang, B., and Liao, Y.: Molecular characterization of organic aerosol in the Himalayas: insight from ultra-high-resolution mass spectrometry, Atmos. Chem. Phys, 19, 1115-1128, 10.5194/acp-19-1115-2019, 2019.*

*Chauveau-Duriot, B., Doreau, M., Noziere, P., and Graulet, B.: Simultaneous quantification of carotenoids, retinol, and tocopherols in forages, bovine plasma, and milk: validation of a novel UPLC method, Analytical and Bioanalytical Chemistry, 397, 777-790, 10.1007/s00216-012-6610-6, 2010.*

*Chen, Y., Bond, T.C.: Light absorption by organic carbon from wood combustion. Atmos. Chem. Phys. 10, 1773−1787, 10.5194/acp-10-1773-2010, 2010.*

*Chen, Q., Mu, Z., Song, W., Wang, Y., Yang, Z., Zhang, L., Zhang, Y.L.: Size‐resolved characterization of the chromophores in atmospheric particulate matter from a typical coal‐burning city in China. J. Geophys. Res.: Atmos., 124, 10546-10563, 10.1029/2019JD031149, 2019.*

*Cheng, Y., He, K.B., Du, Z.Y., Engling, G., Liu, J.M., Ma, Y.L., Zheng, M., Weber, R.J.: The characteristics of brown carbon aerosol during winter in Beijing. Atmos. Environ., 127, 355–364, 10.1016/j.atmosenv.2015.12.035, 2016.*

*Kujawinski, E.: Electrospray ionization Fourier transform ion cyclotron resonance mass spectrometry (ESI FT-ICR MS): characterization of complex environmental mixtures. Environmental Forensics, 3(3–4), 207–216. 10.1006/enfo.2002.0109, 2002.*

*Lin, P., Fleming, L.T., Nizkorodov, S.A., Laskin, J., Laskin, A.: Comprehensive molecular characterization of atmospheric brown carbon by high resolution mass spectrometry with electrospray and atmospheric pressure photoionization. Anal. Chem., 90, 12493-12502, 10.1021/acs.analchem.8b02177, 2018.*

*Liu, J., Bergin, M., Guo, H., King, L., Kotra, N., Edgerton, E., and Weber, R. J.: Size-resolved measurements of brown carbon in water and methanol extracts and estimates of their contribution*

*to ambient fine-particle light absorption, Atmos. Chem. Phys., 13, 12389–12404, 10.5194/acp-13-12389-2013, 2013.*

*Shen, Z.; Zhang, Q.; Cao, J.; Zhang, L.; Lei, Y.; Huang, Y.; Huang, R. J.; Gao, J.; Zhao, Z.; Zhu, C.; Yin, X.; Zheng, C.; Xu, H.; Liu, S.: Optical properties and possible sources of brown carbon in PM2.5 over Xi'an, China. Atmos. Environ., 150, 322−330, 10.1016/j.atmosenv.2016.11.024, 2017.*

*Zeng, Y., Ning, Y., Shen, Z., Zhang, L., Zhang, T., Lei, Y., Zhang, Q., Li, G., Xu, H., Ho, S. S. H., and Cao, J.: The roles of N, S, and O in molecular absorption features of brown carbon in pm$_{2.5}$ in a typical semi‑arid megacity in northwestern China, J. Geophys. Res.: Atmos., 126, 10.1029/2021JD034791, 2021.*

*Zeng, Y., Shen, Z., Takahama, S., Zhang, L., Zhang, T., Lei, Y., Zhang, Q., Xu, H., Ning, Y., Huang, Y., Cao, J., and Rudolf, H.: Molecular absorption and evolution mechanisms of PM$_{2.5}$ brown carbon revealed by electrospray ionization Fourier transform–ion cyclotron resonance mass spectrometry during a severe winter pollution episode in Xi'an, China, Geophys. Res. Lett., 47, 10.1029/2020GL087977, 2020.*

*Wang, X., Hayeck, N., Brüggemann, M., Yao, L., Chen, H., Zhang, C., Emmelin, C., Chen, J., George, C., and Wang, L.: Chemical characteristics of organic aerosols in Shanghai: A study by ultrahigh-performance liquid chromatography coupled with orbitrap mass spectrometry, J. Geophys. Res.: Atmos., 122, 11,703-711,722, 10.1002/2017jd026930, 2017.*

Quantitation

ESI is not a very quantitative method, as the ionization efficiency of molecules varies drastically between compounds due to size and functional group.1 However, the authors employ relative intensity - which I don't think is clearly defined (see minor comment) - as the sole indicator for the abundance and concentration of species in the sample.

**Response**: In our study, the "intensity" for each molecule represents ion signal response, which corresponds to the concentration multiplied by its ionization efficiency and can be measured directly from the UHPLC-Q-TOF MS/MS instrument (Xu et al., 2018; Zeng et al., 2020). Meanwhile, the "relative intensity" presented in Table 2 and Section 3.3 is expressed as a percentage of each molecule's ion intensity for the sum ion

intensity of the corresponding molecular subgroups, which made a repeated concept with "fraction".

To avoid misunderstanding, the "relative intensity" was revised as "ion intensity" in the text and the detailed corrections were listed as a response to the minor comment as below. Meanwhile, the definition of "intensity" was added in the revised manuscript as follows.

*Lines 123-124:*

*"During the detection, the ion intensity refers to ion signal response and mass-to-charge ratio (m/z) were both obtained."*

Certainly, as the reviewer mentioned that the intensity measured from ESI-UHPLC-Q-TOF MS/MS cannot be treated as an indicator of the abundance and concentration of species. The concentration of molecules in the sample was hardly obtained and was not discussed in our study. However, it's noted that BrC is a kind of significant light-absorbing organic carbon. These molecular absorption data of each molecule were more favorable to discussing BrC's absorption and environmental influences than concentration. As depicted in our study, the ion intensity is a key parameter to calculate molecular absorptions using partial least squares regression (PLSR). Thus, in our study, combing with the light absorption coefficient ($b_{abs}$) of each sample, the intensity data and the PLSR method, the level of molecular absorption of each molecule in the sample was obtained.

References:

Xu, Y., Cai, H., Cao, G., Duan, Y., Pei, K., Tu, S., Zhou, J., Xie, L., Sun, D., and Zhao, J.: Profiling and analysis of multiple constituents in Baizhu Shaoyao San before and after processing by stir-frying using UHPLC/Q-TOF-MS/MS coupled with multivariate statistical analysis, J. Chromatogr. B, 1083, 110-123, https://doi.org/10.1016/j.jchromb.2018.03.003, 2018.

Zeng, Y., Shen, Z., Takahama, S., Zhang, L., Zhang, T., Lei, Y., Zhang, Q., Xu, H., Ning, Y., Huang, Y., Cao, J., and Rudolf, H.: Molecular absorption and evolution mechanisms of $PM_{2.5}$ brown carbon revealed by electrospray ionization fourier transform–ion cyclotron resonance mass spectrometry during a severe winter pollution episode in Xi'an, China, Geophys. Res. Lett., 47, https://doi.org/10.1029/2020GL087977, 2020.

Identification

Molecular identification using MS, even with the assistance of MS/MS, is challenging. How can the authors be confident about structure identification (Table 2) down to the isomer level? As the authors stated themselves, there are numerous isomers in this complex sample (Line 206). It seems that the authors have investigated a lot of work into a self-built library (Line 113), but it is not clear in the manuscript.

**Response**: Yes, the MS/MS method are hardly identifying the molecular structure from thousands of molecules. Previous studies have confirmed that MS-MS spectra can truly deduce structural information with the target mode, such as NACs detection (Xie et al., 2020; Zhang et al., 2022). However, it's truly hard to distinguish the compounds' structure under non-target mode because plenty of fragment ions appeared in the same retention time, especially for isomers. Actually, based on the $ESI^-$ UHPLC-Q-ToF MS/MS determination and PLSR simulation, the related descriptions of Table 2 in the main text mainly focus on the determination of molecular formulae including double-bond equivalence (DBE) and aromatic equivalent (Xc), the level of molecular absorption coefficient, intensity, and their percentages. Meanwhile, the basis for assuming the specific molecular structure of BrC (i.e., $C_{20}H_{26}O_2N_2$, $C_{26}H_{24}ON_4S$, and $C_{22}H_{32}O_2$) in Section 3.3 was their elemental constituents and carbon bonded types. As supported, these assumptions were often seen in the exploration of MS instruments, such as the FTICR-MS in the study of Tang et al. (2020). Consequently, the key objective of our work is to ensure the accuracy of BrC's molecular formulae identification rather than their structure recognition. After careful consideration, we have decided to delete the "*possible name*" and "*potential structure*" in Table 2 and added the processes of molecular formulae identification in detail to the revised manuscript.

The self-built library in our work contains two parts: one is the original library in the platform of UNIFI software v1.9.4 (Waters Corp., Milford, MA, USA) in $ESI^-$ UHPLC-Q-ToF MS/MS; the other is the natural product library of National Institute of Standards and Technology (NIST). Considering to the leading role of NIST (National Institute of Standards and Technology) in the development of mass spectrum database

and the retrieval of mass spectrum data (Lowenthal et al., 2013), our study has selected the natural product library of NIST to directly complements current database by providing more organic carbonaceous compounds. The natural product library of NIST mainly contains fatty acid compounds, flavonoids, coumarin derivatives, terpenoids, carbohydrate compounds, furan compounds, aromatic alcohols, phenols, etc.

As mentioned above, we have reinforced the descriptions of molecular formulae identification and self-built library in the revised manuscript as below.

*Lines 124-133:*

*"During the detection, the ion intensity refers to ion signal response and mass-to-charge ratio (m/z) were both obtained. Both m/z data and ion intensities were processed on the platform of the UNIFI Software 1.9.4 (Waters Corp., Milford, MA, USA) to assign the possible molecular formulae. The lower and upper limitations for a peak intensity of energy detection were set as 80 and 200, respectively. The mass error for the molecular formula assignment did not exceed ±2 mDa. To ensure the accuracy of MSOC molecular formulae identification, a three-step scheme was built. First, the negative ESI UHPLC-Q-TOF MS/MS data sets were preliminarily processed using the platform UNIFI for chromatographic peak deconvolution, data normalization, and quality calibration. Second, the original library in UNIFI software was combined with the library of the National Institute of Standards and Technology (NIST) to establish the self-built library. Third, both the measured data sets of MSOC compounds and the database constructed in the self-built library were imported into UNIFI software, and thus the background subtraction and formula assignment were performed (Xu et al., 2020b; Mardal et al., 2018)."*

In addition, the speculations of the molecular structure were rephrased as follows in the revised manuscript.

*Lines 303-304:*

*"The $C_{26}H_{24}ON_4S$ compound hypothetically contains 4-5 benzene rings and a carbonyl group with the highest unsaturation"*

*Lines 317-319:*

*"In contrast, the compounds $C_{18}H_{18}O_4$, $C_{19}H_{22}O_3$, and $C_{22}H_{32}O_2$ with relatively high DBE and Xc values potentially containing one or two aromatic rings and -COOH are regarded as unique markers for dung combustion (Graham et al., 2002)."*

*Lines 352-353:*

*"On the other hand, the unique CHO markers for the dung combustion can be identified as $C_{18}H_{18}O_4$, $C_{19}H_{22}O_3$, and $C_{22}H_{32}O_2$, which consist of aromatic rings and -COOH groups."*

References:

Graham, B., Mayol-Bracero, O. L., Guyon, P., Roberts, G. C., Decesari, S., Facchini, M. C., Artaxo, P., Maenhaut, W., Köll, P., and Andreae, M. O.: Water-soluble organic compounds in biomass burning aerosols over Amazonia 1. Characterization by NMR and GC-MS, J. Geophys. Res.: Atmos., 107, LBA 14-11-LBA 14-16, https://doi.org/10.1029/2001JD000336, 2002.

Lowenthal, M. S., Andriamaharavo, N. R., Stein, S. E., and Phinney, K. W.: Characterizing vaccinium berry standard reference materials by GC‑MS using NIST spectral libraries, Analytical and bioanalytical chemistry, 405, 4467-4476, 10.1007/s00216-012-6610-6, 2013.

Mardal, M., Dalsgaard, P. W., Qi, B., Mollerup, C. B., Annaert, P., and Linnet, K.:Metabolism of the synthetic cannabinoids AMB-CHMICA and 5C-AKB48 in pooled human hepatocytes and rat hepatocytes analyzed by UHPLC-(IMS)-HR-MSE, J. Chromatogr. B, 1083, 189-197, 10.1016/j.jchromb.2018.03.016, 2018.

Tang, J., Li, J., Su, T., Han, Y., Mo, Y., Jiang, H., Cui, M., Jiang, B., Chen, Y., Tang, J., Song, J., Peng, P. A., and Zhang, G.: Molecular compositions and optical properties of dissolved brown carbon in biomass burning, coal combustion, and vehicle emission aerosols illuminated by excitation–emission matrix spectroscopy and Fourier transform ion cyclotron resonance mass spectrometry analysis, Atmos. Chem. Phys., 20, 2513-2532, https://doi.org/10.5194/acp-20-2513-2020, 2020.

Xie, M., Zhao, Z., Holder, A. L., Hays, M. D., Chen, X., Shen, G., Wang, Q. G.: Chemical composition, structures, and light absorption of N-containing aromatic compounds emitted from burning wood and charcoal in household cookstoves. Atmos. Chem. Phys, 20(22), 14077-14090, 10.5194/acp-20-14077-2020, 2020.

*Xu, L., Liu, Y., Wu, H., Wu, H., Liu, X., and Zhou, A.:Rapid identification of chemical profile in Gandou decoction by UPLC-Q-TOF-MSE coupled with novel informatics UNIFI platform, J. Pharm. Anal., 10(1), 35-48, 10.1016/j.jpha.2019.05.003, 2020.*

*Zhang, Q., Li, Z., Shen, Z., Zhang, T., Zhang, Y., Sun, J., Zeng, Y., Xu, H., Wang, Q., and Ho, S. S. H.: Source profiles of molecular structure and light absorption of PM$_{2.5}$ brown carbon from residential coal combustion emission in Northwestern China, Environ. Pollut., 299, 118866, 10.1016/j.envpol.2022.118866, 2022b.*

Combining completeness, quantitation, and identification above, many of the authors' conclusions and implications are questionable and should be revised across the manuscript. Some of these include, but may not be limited to:

- The Mbabs values and the relative contribution of each compound class to the total absorption (basically everything in Figure 6).

**Response**: The *Mb$_{abs}$* value represents the molecular mass absorption coefficient for individual molecular formulas. Following our previous studies (Zhang et al., 2022; Zeng et al., 2020), the *Mb$_{abs}$* for each molecule corresponds to the ion intensity multiplied by its calibration coefficient (β). In response to the comment above, the "intensity" refers to the ion signal response that can be directly detected from ESI⁻ UHPLC-Q-TOF MS/MS. The "β" is an integrated reflection of UV–Vis absorption and ionization efficiency for individual molecules. The PLSR was designed to process many predictor variables (potentially correlated with each other) with a limited sample size, which was used here to determine how closely mass spectral intensity and BrC absorption were related to each other. Therefore, combing the information of 16 BrC b$_{abs365}$ data, hundreds of molecular formulae, and their ion intensities, the "β" can be predicted using PLSR analysis. Finally, the *Mb$_{abs}$* value of individual molecules was obtained.

To express the determination of *Mb$_{abs}$* explicitly, we have added these sentences to Section 2.4 in the revised manuscript as below.

*Lines 137-144:*

*"To determine the relationship between the MSOC b$_{abs365}$ and their detected molecules, the molecular absorption coefficient (Mb$_{abs}$), which represents the light-*

*absorbing coefficient of individual MSOC molecules at 365 nm, was calculated. Following the steps applied in our previous studies, the $Mb_{abs}$ for each molecule corresponding to the ion intensity were multiplied by its calibration coefficient (β). The "β" is an integrated reflection of UV–vis absorption and ionization efficiency for individual molecules, which can be determined from the combination of 16 MSOC $b_{abs365}$ data, hundreds of detected molecular formulas, and their ion intensities using partial least squares regression (PLSR) analysis (Mehmood et al., 2019; Rambo et al., 2016; Zeng et al., 2020)."*

Meanwhile, the "total absorption" shown in Figure 6 refers to the sum of the molecular absorption coefficient of measured each compound group including CHO, CHON, and CHONS with negative ESI sources. Further, "their relative contribution" in Figure 6 means the percentages of $Mb_{abs}$ for CHO, CHON, and CHON compounds in the total absorption in the six combustion scenarios. To clarify it more precisely, the caption of Figure 6 has been revised as below.

*"Figure 6. Comparison of (a) the molecular mass absorption coefficient ($Mb_{abs}$), and (b) the relative contributions of $Mb_{abs}$ of each molecular subgroup to the total $Mb_{abs}$ assigned in negative ESI ionization modes from six combustion scenarios. The total $Mb_{abs}$ presented here refers to the sum of $Mb_{abs}$ for CHO, CHON, and CHONS groups with negative ESI source detection."*

References:

*Mehmood, T., Sadiq, M., and Aslam, M.: Filter-based factor selection methods in partial least squares regression, IEEE Access, 7, 153499-153508, 10.1109/access.2019.2948782, 2019.*

*Rambo, M. K. D., Ferreira, M. M. C., and Amorim, E. P.: Multi-product calibration models using NIR spectroscopy, Chemometr. Intell. Lab. Sys., 151, 108-114, 10.1016/j.chemolab.2015.12.013, 2016.*

*Zeng, Y., Shen, Z., Takahama, S., Zhang, L., Zhang, T., Lei, Y., Zhang, Q., Xu, H., Ning, Y., Huang, Y., Cao, J., and Rudolf, H.: Molecular absorption and evolution mechanisms of $PM_{2.5}$ brown carbon revealed by electrospray ionization Fourier transform–ion cyclotron resonance mass spectrometry during a severe winter pollution episode in Xi'an, China, Geophys. Res. Lett., 47, 10.1029/2020GL087977, 2020.*

*Zhang, Q., Li, Z., Shen, Z., Zhang, T., Zhang, Y., Sun, J., Zeng, Y., Xu, H., Wang, Q., and Ho, S. S. H.: Source profiles of molecular structure and light absorption of $PM_{2.5}$ brown carbon from residential coal combustion emission*

*in Northwestern China, Environ. Pollut., 299, 118866, 10.1016/j.envpol.2022.118866, 2022b.*

- The fact that authors consider the observed BrC species are all that contribute to light absorbers.   E.g., "The values confirm that CHON compounds were likely to be the dominant light absorbers in solid fuel combustion over the TPL region (Line 254)".

**Response**: There are two approaches to identifying light-absorbing BrC species in our study.

Firstly, hundreds of molecules in MSOC were measured by negative ESI UHPLC-Q-TOF MS/MS instrument. Thus, the MSOC extracts discussed in our study only focus on the polar components. In response to the above comment, the limitations that existed during the processes of both MSOC extraction and ESI detection were supplied in the revised manuscript.

Secondly, based on the obtained $Mb_{abs}$ data of individual MSOC molecules using PLSR analysis, the light-absorbing MSOC molecules were selected and can be used as a surrogate for the behavior of BrC constituents. It's noted that the value of $Mb_{abs}$ is a unique basis to determine whether the MSOC molecule is BrC (Zeng et al., 2021). In this study, the MSOC molecules with high $Mb_{abs}$ ($\geq 10^{-8}$) were selected as BrC while $Mb_{abs} < 10^{-8}$ as non-BrC molecules. Non-BrC molecules are molecules with negative or extremely low absorptions, which were not discussed in this paper because of their negligible absorption contributions. Thus, the selection of BrC molecules according to $Mb_{abs}$ data was added to Section 2.3 in the revised manuscript as below.

*Lines 149-151:*

*"Furthermore, the molecules with high $Mb_{abs}$ values ($\geq 10^{-8}$) were selected as BrC molecules, while $Mb_{abs} < 10^{-8}$ as non-BrC molecules. After these selections, the $Mb_{abs}$ of CHO, CHON, and CHONS groups would be identified as the light-absorbing BrC molecules."*

Meanwhile, the related sentences of subgroup compounds $Mb_{abs}$ mentioned in the text was revised as follows:

*Lines 283-284:*

*"The values confirm that the CHON compounds are the most dominant light*

*absorbers among these detected light-absorbing BrC molecules in solid fuel combustion in the TPL region.*"

*Lines 286-287:*

"*the sum of CHO and CHON contributes up to 99.7% of the total detected BrC $Mb_{abs}$ in the sheep and yak dung combustions.*"

*Lines 288-290:*

"*. In contrast, the simulated $Mb_{abs}$ values of CHONS compounds show low fractions in the total detected BrC $Mb_{abs}$, which are only 16.5 and 26.6% of the values for the cooking and heating combustion of bituminous coal, respectively.*"

References:

*Zeng, Y., Ning, Y., Shen, Z., Zhang, L., Zhang, T., Lei, Y., Zhang, Q., Li, G., Xu, H., Ho, S. S. H., and Cao, J.: The Roles of N, S, and O in molecular absorption features of brown carbon in $PM_{2.5}$ in a typical semi-arid megacity in northwestern China, J.Geophys. Res.: Atmos., 126, https://doi.org/10.1029/2021JD034791, 2021.*

- The conclusion "this work provides exhaustive molecular information" (Line 327)

**Response**: To express it more precisely, both the limitation of molecular detection and further study schedule was added to the revised conclusion as below.

*Line 353-357:*

"*Therefore, under the various dung fuels and stove combinations with oxygen-deficient combustion, our results provide exhaustive molecular information on the polar BrC emission inventory in negative ESI mode. This could better elaborate the high polar BrC emissions and their radiative efficiency. Considering that ESI is limited to the detection of polar BrC molecules, a reliable molecular characteristic-based method should be developed to examine the nonpolar or less polar BrC molecules in the future.*"

Burning conditions/scenarios.

One of the conclusions from the paper is that burning type/scenario (i.e., cooking vs heating) significantly affects BrC emissions. The conclusion is valid and interesting, but the authors did not provide further information on how they are different. The readers can only see that they are different, their pictures (SI) and "heating has poor oxygen conditions than cooking" (Line 152), but without further support. I thought the authors measured OC/EC to gauge combustion efficiency, but I was surprised that none of the data had been presented in the manuscript.

**Response**: Thanks for your comment. The modified combustion efficiency (MCE), defined as $CO_2/(CO_2+CO)$, was calculated to judge the combustion state of flaming- and smoldering during the burn (McMeeking et al., 2009; Shen et al., 2013) and thus it is worth to present. Therefore, we have added the emission factors (EFs) of $CO_2$, $CO$, $O_2$ concentration, and MCE value in Table S2 in the revised manuscript to explicitly describe how we use MCE to judge the combustion state. Jen et al. (2019) have confirmed that the MCE values near 1 indicate almost pure flaming, while values near 0.8 are almost pure smoldering, with 0.9 representing a roughly equal mix of these processes. In this study, the burning conditions varied for all combustion scenarios, with the MCE ranging from 0.90 to 0.99, reflecting the mix of flaming and smoldering combustion. The average MCE observed in the heating scenarios was 0.94, significantly lower than those in the cooking scenarios (0.97; $p < 0.05$). Meanwhile, the excess air ratio (defined as the ratio of the actual air quantity to theoretical air requirement) supplied in Table S2 was also high (1.21-1.44) in cooking combustion, while decreasing by 7.3-11.3% in heating combustion. The above results confirmed that the fuels in heating combustion suffered from a serious deficient air supply, which led to incomplete combustion occurred frequently than those in cooking combustion. In addition, the range of OC/EC ratio reported in the study is very large and displayed the inverse trend with MCE, even for the same fuels, better supporting that smoldering and incomplete combustion frequently occurred in heating scenarios (Wang et al., 2020; McMeeking et al., 2014; Pokhrel et al., 2016).

To clarify how the burning scenario (i.e., cooking vs heating) significantly affects BrC emissions, we have supplied the measurements of $CO_2$, $CO$, and $O_2$ concentrations

in the revised Section 2.1. Also, the discussions of MCE, $O_2$ supply, and OC/EC ratio were added in the revised manuscript. The related contents are shown below.

*Lines 91-94:*

*"Furthermore, the online concentrations of carbon dioxide ($CO_2$) (range: 1-10000 ppm) and carbon monoxide (CO) (range: 1-1000 ppm) in the diluted smoke were monitored using infrared-based sensors (Sundo Technology, Shenzhen, China). Oxygen ($O_2$) concentration was measured by an electrochemical sensor (Sundo Technology, Shenzhen, China) with a measurement range of 0-25%."*

*Lines 179-186:*

*"Considering the burning conditions shown in Table S4, the modified combustion efficiency [defined as $CO_2= CO_2 / (CO_2+CO)$] on the combustion is generally low and displays a roughly equal mix of smoldering and flaming combustion, whereas the MCE increased significantly in the cooking indicates pure flaming combustion (Jen et al., 2019; Wang et al., 2020b). Meanwhile, the excess air ratio, defined as the ratio of the actual air quantity to theoretical air requirement, shows high values (1.21-1.44) in the cooking combustion, while a 7.3-11.3% lower in the heating. These results indicate that the fuel combustion for heating suffered from a poor oxygen supply and existed in the smoldering process, leading to abundant OC emissions, high OC/EC ratio, and stronger formation of light-absorbing BrC molecules than those in the cooking scenario (McMeeking et al., 2009; Sun et al., 2021a; Shen et al., 2013)."*

Table S4 EFs of CO and $CO_2$, $O_2$ concentrations, MCE and OC/EC ratio from different combustion scenarios.

| Combustion scenario | | $O_2$, % | Excess air ratio | CO, $g \cdot kg^{-1}$ | $CO_2$, $g \cdot kg^{-1}$ | MCE, % | OC/EC |
|---|---|---|---|---|---|---|---|
| Yak dung cooking | Mean | 3.56 | 1.21 | 3.70 | 227.04 | 0.97 | 4.27 |
| | Sd | 0.54 | 0.04 | 1.45 | 35.10 | 0.01 | 1.05 |
| Yak dung heating | Mean | 2.22 | 1.12 | 25.76 | 744.27 | 0.94 | 7.48 |
| | Sd | 0.08 | 0.01 | 6.65 | 248.57 | 0.01 | 2.26 |

| | | | | | | | |
|---|---|---|---|---|---|---|---|
| Sheep dung cooking | Mean | 6.32 | 1.44 | 8.11 | 285.51 | 0.95 | 7.73 |
| | Sd | 0.65 | 0.06 | 2.15 | 91.32 | 0.00 | 3.10 |
| Sheep dung heating | Mean | 4.43 | 1.27 | 17.53 | 419.83 | 0.93 | 20.45 |
| | Sd | 1.01 | 0.08 | 3.46 | 75.02 | 0.02 | 10.89 |
| Bitumite cooking | Mean | 4.06 | 1.25 | 14.32 | 1303.70 | 0.98 | 28.78 |
| | Sd | 1.45 | 0.11 | 6.66 | 254.10 | 0.01 | 9.92 |
| Bitumite heating | Mean | 2.04 | 1.11 | 33.43 | 1157.67 | 0.95 | 173.29 |
| | Sd | 0.65 | 0.04 | 3.95 | 215.97 | 0.02 | 97.11 |

*Lines 195-197:*

*"These comparisons suggest that the poor oxygen supply (excess air ratio<2) and stacking conditions lead to incomplete dung combustion, resulting in the release of substantial amounts of strong light-absorbing BrC into TPL atmospheric aerosols (Huo et al., 2018). "*

*Lines 303-304:*

*"The results indicate that the flaming in cooking combustion with a relatively high oxygen supply favors the oxidized BrC compound formation."*

*Line 333-335:*

*"The most probable reason for the variance is that these fuels produce much more abundant BrCs with relatively stronger light-absorbing capacities in cooking than in heating combustions."*

*Lines 336-341:*

*"It should be noted that the emission of bitumite fuels displays a higher level of OC/EC ratio (Table S4) than that in the two dung fuels. However, as indicated in Figures S1c and d, the combustions of the dungs could produce more CHON and CHO compounds with high aromaticity and $Mb_{abs}$ during oxygen-deficient combustion,*

*consequently yielding much higher BrC SFE (Zhang et al., 2022a; Sun et al., 2021a). Therefore, both the fuels and combustion scenarios must be taken into consideration for minimizing the emissions of atmospheric BrC and their radiative forcing influences in the plateau region."*

References:

Huo, Y., Li, M., Jiang, M., and Qi, W.: Light absorption properties of HULIS in primary particulate matter produced by crop straw combustion under different moisture contents and stacking modes, Atmos. Environ., 191, 490-499, https://doi.org/10.1016/j.atmosenv.2018.08.038, 2018.

Jen, C. N., Hatch, L. E., Selimovic, V., Yokelson, R. J., Weber, R., Fernandez, A. E., Kreisberg, N. M., Barsanti, K. C., and Goldstein, A. H.: Speciated and total emission factors of particulate organics from burning western US wildland fuels and their dependence on combustion efficiency, Atmos. Chem. Phys., 19(2), 1013-1026, 10.5194/acp-19-1013-2019, 2019.

McMeeking, G. R., Kreidenweis, S. M., Baker, S., Carrico, C. M., Chow, J. C., Collett Jr, J. L., and Wold, C. E.: Emissions of trace gases and aerosols during the open combustion of biomass in the laboratory. J. Geophys. Res. Atmos., 114(D19), 10.1029/2009JD011836, 2009.

McMeeking, G., Fortner, E., Onasch, T., Taylor, J., Flynn, M., Coe, H., and Kreidenweis, S.: Impacts of nonrefractory material on light absorption by aerosols emitted from biomass burning, J. Geophys. Res. Atmos., 119, 12,272-212,286, 2014.

Pokhrel, R. P., Wagner, N. L., Langridge, J. M., Lack, D. A., Jayarathne, T., Stone, E. A., Stockwell, C. E., Yokelson, R. J., and Murphy, S. M.: Parameterization of single-scattering albedo (SSA) and absorption Ångström exponent (AAE) with EC/OC for aerosol emissions from biomass burning, Atmos. Chem. Phys., 16, 9549-603 9561, 10.5194/acp-16-9549-2016, 2016.

Sun, J., Shen, Z., Zhang, B., Zhang, L., Zhang, Y., Zhang, Q., Wang, D., Huang, Y., Liu, S., and Cao, J.: Chemical source profiles of particulate matter and gases emitted from solid fuels for residential cooking and heating scenarios in Qinghai-Tibetan Plateau, Environ. Pollut., 285, 117503, 10.1016/j.envpol.2021.117503, 2021a.

Shen, G., Xue, M., Wei, S., Chen, Y., Wang, B., Wang, R., Shen, H., Li, W., Zhang, Y., Huang, Y., Chen, H., Wei, W., Zhao, Q., Li, B., Wu, H., and Tao, S.: Influence of fuel mass load, oxygen supply and burning rate on emission factor and size distribution of carbonaceous particulate matter from indoor corn straw burning, J. Environ. Sci., 25, 511-519, 10.1016/S1001-0742(12)60191-0, 2013.

Wang, Y., Hu, M., Xu, N., Qin, Y., Wu, Z., Zeng, L., He, L.: Chemical composition and light absorption

of carbonaceous aerosols emitted from crop residue burning: influence of combustion

efficiency. Atmos. Chem. Phys., 20(22), 13721-13734, 10.5194/acp-20-13721-2020, 2020.

Zhang, B., Shen, Z., Sun, J., He, K., Zou, H., Zhang, Q., Li, J., Xu, H., Liu, S., Ho, K. F., and Cao, J.:

County-level of particle and gases emission inventory for animal dung burning in the Qinghai–

Tibetan Plateau, China, J. Cleaner Prod., 367, 10.1016/j.jclepro.2022.133051, 2022a.

**Minor Comments**

- I don't think Mbabs is clearly defined in the main article (it is in the abstract).
As such, the method to determine Mbabs is unclear. Is the PLSR method from line ~118
for Mbabs?

- 'Relative intensity' is not properly defined. Relative to what?

**Response**: Thanks for the reminder. The "intensity" refers to the detected ion response.
The "relative intensity" in the text represents the percentages of individual molecules'
intensity among the sum of the intensity of their corresponding subgroups. Therefore,
the concept "relative" and "fraction" were repeated in the article and the "relative" or
"relatively" in the main text should be deleted. Also, depending on the meaning of our
context, the "relative intensity" presented in the text and Figure S5 should be corrected
as "ion intensity". We have revised these incorrect expressions in the revised
manuscript as follows.

*Line 240:*

"*in terms of both numbers and ion intensities*"

Lines 241-242:

"*Especially, the sum of CHON and CHO compounds account for 94.3-94.6% and
79.4-88.3% of the total number and ion intensity, respectively,*"

Lines 291-292:

*"Among all combustion experiments, the specific CHO, CHON, and CHONS compounds with high ion intensities and simulated $Mb_{abs}$ values are identified and listed in Table 2."*

Line 293:

*"with high fractions of ion intensity"*

Lines 294-295:

*"in terms of ion intensity (>3%) and $Mb_{abs}$ (>3%)"*

Lines 300-301:

*"contribute 52.5-59.5% and 21.9-38.6% to the total ion intensity and $Mb_{abs}$"*

[Figure]

Figure S5. Percentage contributions of individual molecular subgroups to the number of total molecules in different fuel types (left); percentage contributions of individual molecular subgroups to the total absorption ion intensity in different fuel types (right).

Additionally, the "intensity" in Table 2 was corrected as "ion intensity". The explanation of "fraction" was added to the revised Table 2 as below.

*"Fraction here refers to the contribution of $Mb_{abs}$ and ion intensity of individual molecules to the sum of $Mb_{abs}$ and ion intensity of their corresponding subgroups."*

- Line 218 - what is atmospheric conditions? Is it burning conditions instead?

**Response:** Yes, the "*atmospheric condition*" here should be revised as "*burning conditions*".

•     In the paragraph starting Line 220, the discussion related to Figure 4 does not seem to fully reflect the observations on Figure 4. For example, 'a wide DBE range from 0 to 20' - it seems to be wider. And 'CHON and CHONS compounds (beyond 70%) lies in the potential BrC chromophore range - I don't think its beyond 70%, looking at the data.

**Response**: Yes, the expression of Figure 4 is not ideally clear and made a little bit of misunderstanding. We have corrected "*a wide DBE range from 0 to 20*" to "*a wide DBE range from 0 to 40*". Actually, the fraction beyond 70% represents the contribution of CHON and CHONS compounds to the total molecular compounds that lie in the potential BrC chromophores ranges. These results were well depicted in the pie chart in Figure 4. To make this expression more explicit, we have rephrased the sentence and rewrote the Figure 4 caption as follows.

*Lines 254-255:*

*"The CHON and CHONS account for 72.5-79.2% of the total molecules and appear in the potential BrC chromophores range."*

*Figure 4 Caption:*

*"Plot of the double bond equivalent (DBE) vs the number of carbon atoms in identified species detected in six combustion models. Lines indicate DBE reference values of linear conjugated polyenes $C_xH_{x+2}$ (green solid line), data-condensed PAHs (Kelly solid line), and fullerene-like hydrocarbons with DBE = 0.9 × c (orange solid line). Data points inside the blue-shaded area are potential BrC chromophores. The pie chart shows the percentage of CHO (red), CHON (dark blue), and CHONS (earthy yellow) to the total number of molecular compounds in the blue-shaded area.*"

**Grammatical errors**

There are numerous that I believe can be corrected by careful proofreading and editing. Additionally, certain sentences appear too subjective, and I suggest the authors revise the tone of the writing.

E.g, :

"We all know that burning biofuels of yak and sheep dung is still the most traditional and popular way of heating and cooking over the TPL region. The total consumption of both yak and sheep dung can even possess up to nearly 70% of the total dung fuel consumptions during heating periods (Zhang et al., 2022a)."

'We all know that' , 'still the most traditional and' , 'dung can even possess up to' can all be considered subjective.

**Response**: Suggestion taken. Thoroughly polish has been completed in the revised manuscript to avoid subjective expressions and to correct grammatical errors. The sentences dealing with subjective assumptions have been revised as follows.

*Line 36:*

"*which would produce an extreme impact on the raise of ambient air temperature*"

*Lines 56-57:*

"*Meanwhile, researchers began to thoroughly investigate possible BrC molecules using high-performance and resolution mass spectrometric instruments.*"

*Lines 59-60:*

"*Consequently, the current studies would limit the accurate evaluation of the light-absorbing abilities of BrC*"

*Lines 203-204:*

"*which would be mainly attributed to the low-temperature and deficient oxygen burning conditions*"

*Line 209:*

"*probably ascribed to the different constituent molecules*"

*Lines 307-309:*

*"These compounds possessed enough oxygen atoms to allow the assignments of -OSO$_3$H and/or -ONO$_2$ groups in their formulas, which might be regarded as OSs or nitroxyl-OSs"*

*Line 316:*

*"which is selected as an important marker for residential combustion in TPL."*

*Lines 343-344:*

*"Biofuels of yak and sheep dung are one of the most traditional and popular materials used for heating and cooking in the TPL region. Their consumption contributes nearly 70% of the total dung fuel consumption during the heating periods"*

Reference

(1) Kenseth, C. M.; Hafeman, N. J.; Huang, Y.; Dalleska, N. F.; Stoltz, B. M.; Seinfeld, J. H. Synthesis of Carboxylic Acid and Dimer Ester Surrogates to Constrain the Abundance and Distribution of Molecular Products in α-Pinene and β-Pinene Secondary Organic Aerosol. Environ. Sci. Technol. **2020**, 54 (20), 12829–12839. https://doi.org/10.1021/acs.est.0c01566.

**Response**: Thanks. This reference was cited in the revised manuscript.

---

## Author Comment (AC2)

**RC2**: ['Comment on acp-2022-801'](), Anonymous Referee #2, 15 Feb 2023  reply

Zhang et al. conducted a comprehensive analysis on the light absorption of methanol-soluble organic carbon (OC) and its molecular characteristics from residential heating and cooking scenarios using dung and bitumite. The authors found that BrC absorption emission factors were up to 9 times higher for incomplete dung burning than for bitumite combustion. Nitrogen-containing species with high aromaticity and CHO molecules with benzene rings and -COOH are unique markers of dung-fuel BrC. The potential radiative effects of the identified chromophores were also evaluated by calculating the simple forcing efficiency (SFE). Little information on BrC emissions from residential combustion in the Qinghai Tibet Plateau (TPL) is documented in the literature, and the topic of this study is important for connecting BrC emissions, molecular composition, and radiative effects of organic aerosols. However, I agree with the first reviewer that the manuscript needs significant revision before acceptance for publication. Here are my comments.

**Response:** We thank the reviewer's comments on our manuscript. We have considered each point and responded and revised accordingly.

1. The title is a bit misleading. A question might be raised about "what enhances the optical properties and radiative effects of brown carbon from dung combustion? However, the manuscript is mainly about the light absorption and molecular characteristics of BrC from dung combustion.

**Response**: We agreed with the reviewer that our manuscript mainly focus on the light absorption and molecular characteristics of BrC from dung combustion. Therefore, we have revised the title to "*Light absorption and molecular characteristics of molecular-specific brown carbon formed in dung combustion in the Tibetan Plateau, China*".

2. The definition and calculation of the molecular absorption coefficient ($Mb_{abs}$) was not provided in the main text or supplementary information. Therefore, a large part of *Section 3.3* is not understandable.

**Response**: To make the text in Section 3.3 understandable, the definition of molecular absorption coefficient ($Mb_{abs}$) was supplied in the revised manuscript as below.

*Lines 137-139:*

*"To determine the relationship between the MSOC $b_{abs365}$ and their detected molecules, the molecular absorption coefficient ($Mb_{abs}$), which represents the light-absorbing coefficient of individual MSOC molecules at 365 nm, was calculated."*

In addition, in response to the first reviewer's comments, the calculation of molecular absorption coefficient ($Mb_{abs}$) was described in detail and was added to Section 2.4 in the revised manuscript as follows.

*Lines 139-143:*

*"Following the steps applied in our previous studies, the $Mb_{abs}$ for each molecule corresponding to the ion intensity was multiplied by its calibration coefficient ($\beta$). The "$\beta$" is an integrated reflection of UV–vis absorption and ionization efficiency for individual molecules, which can be determined from the combination of 16 MSOC $b_{abs365}$ data, hundreds of detected molecular formulas, and their ion intensities using partial least squares regression (PLSR) analysis (Mehmood et al., 2019; Rambo et al., 2016; Zeng et al., 2020)."*

3. Page 3, lines 67-70. BrC from flame combustion shows higher absorption than that from smoldering combustion.

**Response**: After our careful checking, the cited reference of Xie et al. (2020) suggested that large molecules of BrC compounds probably generated from flame combustion shows high absorption. Therefore, we have revised the incorrect expressions as follows.

*Line 68-71:*

*"Interestingly, Xie et al. (2020) found that the factors of a variety of fuel types, high relative humidity, and low elemental carbon to organic carbon ratio (i.e., a measurement proxy for burning conditions) involved in flaming combustion probably produce abundant high molecule weight N-containing aromatic compounds and can strongly enhance the light absorption ability of biomass burning BrC."*

Reference:

Xie, M., Zhao, Z., Holder, A. L., Hays, M. D., Chen, X., Shen, G., Jetter, J. J., Champion, W. M., and

Wang, Q. G.: Chemical composition, structures, and light absorption of N-containing aromatic

compounds emitted from burning wood and charcoal in household cookstoves, Atmospheric Chem.

Phys., 20(22), 14077-14090, 10.5194/acp-20-14077-2020, 2020.

4. Section 2.2 and Appendix II section 2.2. It seems that the $MAC_\lambda$ was calculated by dividing $b_{abs}$ by OC concentrations without considering dissolution ability. Since methanol cannot extract all organic materials in particles, the method used in this work may underestimate the $MAC_\lambda$ value. This should be mentioned and discussed in the last paragraph of Section 3 (Page 7, lines 192-195).

**Response**: Owing to that for methanol extracts the use of an organic solvent prohibits determining carbon mass, the MSOC was not directly quantified. As the reviewer's comment, the loss of OC during the methanol extraction process has truly existed. Both Chen et al. (2019) and Cheng et al. (2016) have found that the average MSOC mass accounted for 88% and 85% of the total OC mass of near source and ambient OC in China. In response to the first major comment raised by the first reviewer, the discussion of methanol extraction error was mentioned in new Lines 99-102 in the revised manuscript. Also, the discussion of MAE (namely $MAC_\lambda$) uncertainties in the revised supplementary materials is as follows.

Supplementary lines 69-74:

*"The $b_{abs}$ is the light absorption coefficient of methanol-soluble BrC ($Mm^{-1}$ or $10^{-6}$ $m^{-1}$), and OC represents the thermal-OC filter-based concentration which was measured using a Sunset thermal/optical carbon analyzer ($\mu g \cdot mL^{-1}$). In this study, we assumed that OC was completely dissolved during the methanol extraction processes. However, this should have a limitation on the MAE calculation, as previous studies showed that the loss of OC potentially existed within 15% (Chen et al., 2019; Cheng et al., 2016; Zhang et al., 2022). Consequently, the calculated MAE of MSOC would be underestimated."*

Also, both the underestimation of MAE and their related descriptions were rephrased in Section 3 in the revised manuscript as below.

*Lines 227-229:*

*"As depicted in Section 2.2 in SI, the underestimation of MAE within 15% might exist in the methanol extracts, thus the primary BrC emissions detected in this study exhibit comparable and even lower MAE values than the mixed primary and secondary BrC polluted urban areas…".*

*Lines 232-235:*

*"However, most average BrC MAE and AAE values for ambient aerosols over TPL regions are in relatively low concentration levels (MAE: 0.34-0.77 $m^2 \cdot g^{-1}$; AAE 3.8-6.24) (Kirillova et al., 2016; Wu et al., 2020; Zhang et al., 2017a), suggesting that residential combustion of dung and coal combustion could be an important source of BrC in the atmosphere over the TPL regions."*

References:

*Chen, D., Zhao, Y., Lyu, R., Wu, R., Dai, L., Zhao, Y., Chen, F., Guan, M.: Seasonal and spatial variations of optical properties of light absorbing carbon and its influencing factors in a typical polluted city in Yangtze River Delta, China. Atmos. Environ., 199, 45-54, 10.1016/j.atmosenv.2018.11.022, 2019.*

*Cheng, Y., He, K.B., Du, Z.Y., Engling, G., Liu, J.M., Ma, Y.L., Zheng, M., Weber, R.J.: The characteristics of brown carbon aerosol during winter in Beijing. Atmos. Environ., 127, 355–364, 10.1016/j.atmosenv.2015.12.035, 2016.*

*Kirillova, E. N., Marinoni, A., Bonasoni, P., Vuillermoz, E., Facchini, M. C., Fuzzi, S., and Decesari, S.: Light absorption properties of brown carbon in the high Himalayas, J. Geophy. Res.: Atmos., 121, 9621-9639, 10.1002/2016JD025030, 2016.*

*Wu, G., Wan, X., Ram, K., Li, P., Liu, B., Yin, Y., Fu, P., Loewen, M., Gao, S., and Kang, S.: Light absorption, fluorescence properties and sources of brown carbon aerosols in the Southeast Tibetan Plateau, Environ. Pollut., 257, 113616, 10.1016/j.envpol.2019.113616, 2020.*

*Zhang, Q., Li, Z., Shen, Z., Zhang, T., Zhang, Y., Sun, J., Zeng, Y., Xu, H., Wang, Q., and Ho, S. S. H.: Source profiles of molecular structure and light absorption of $PM_{2.5}$ brown carbon from residential coal combustion emission in Northwestern China, Environ. Pollut., 299, 118866, 10.1016/j.envpol.2022.118866, 2022b.*

*Zhang, Y., Xu, J., Shi, J., Xie, C., Ge, X., Wang, J., Kang, S., and Zhang, Q.: Light absorption by water-*

*soluble organic carbon in atmospheric fine particles in the central Tibetan Plateau, Environ. Sci.*

*Pollut. Res., 24, 21386-21397, 10.1007/s11356-017-9688-8, 2017a.*

5. Section 3.2, lines 208-211. How did the authors determine the relative intensity of individual groups of compounds? Please provide this information in the methods section or in the supplementary information.

**Response**: In fact, the "relative intensity" concept is an inappropriate expression in the original manuscript. The majority of "relative intensity" in our original manuscript was an inappropriate expression. In response to the second minor comment raised by the first reviewer, the "relative intensity" has been corrected to "ion intensity" in the revised manuscript. The definition of intensity and their assignments for individual molecules were supplied in the revised manuscript as follows:

*Lines 123-126:*

*"During the detection, the ion intensity refers to ion signal response and mass-to-charge ratio (m/z) were both obtained. Both m/z data and ion intensities were processed on the platform of the UNIFI Software 1.9.4 (Waters Corp., Milford, MA, USA) to assign the possible molecular formulae. The lower and upper limitations for a peak intensity of energy detection were set as 80 and 200, respectively."*

6. Section 3.3. I suspect that the $Mb_{abs}$ could be simulated by fitting $b_{abs365}$ of aerosol extracts to the identified molecules statistically. Do all the identified CHO, CHON, and CHONS compounds have light absorption at 365 nm? How did the authors quantify individual groups of molecules based on intensity only?

Additionally, UHPLC-Q-ToF MS/MS operated in the ESI⁻ mode cannot identify all organic compounds in methanol extracts of aerosols. ESI is a soft ionization technique, and many organic compounds cannot be ionized and detected, particularly for large molecules. These limitations should be mentioned and discussed.

**Response**: In fact, the $Mb_{abs}$ for each detected molecule were equal to the ion intensity multiplied by its calibration coefficient (β). Among them, the data of "ion intensity" was directly measured by UHPLC-Q-Tof MS/MS while the "calibration coefficient (β)"

value was obtained from fitting $b_{abs365}$ of aerosol extracts to the identified molecules statistically using the PLSR method. As noted in previous studies, the multivariate regression technique of PLSR can handle high-dimensional data and model complex variable relations (Wehrens and Mevik, 2007; Sujaritha et al., 2019), which is thus suitable for the present work with 16 aerosol extracts and thousands of molecules to obtain β. Therefore, the $Mb_{abs}$ of individual molecules can be calculated from the PLSR method.

To ensure the accuracy and completeness of the $Mb_{abs}$ calculation, three measures including MSOC molecular formula assignments, PLSR model optimization, and BrC compounds selection, were conducted in our study.

① MSOC molecular formula assignments. To quantify individual groups of molecules, two datasets of ion intensity and m/z for each MSOC sample were used. The UNIFI software was used to calibrate and assign the possible molecular formulae for all peaks in the selected mass spectra of the active analysis and has been widely attempted in related studies (Xu et al., 2020; Man et al., 2021). The formulae of all measured m/z values could be calculated through the corresponding mass spectra data. The mass error for the molecular formulae assignment did not exceed ±2 mDa. Therefore, the software can be used independently to obtain molecular formula with high precision. The formula and the corresponding ion intensity for each molecule were acquired from the platform UNIFI software. In response to the 3[rd] major comment raised by the first reviewer, the data analysis of the UNIFI platform was added to the new lines 124-134 in the revised manuscript.

② PLSR model optimization. It's noted that the determination of the optimal number of components is very critical for the PLSR. In our study, the 5-fold cross-validation method was used. The series of average statistical parameters between the calculation and validation set includes the coefficient of determination for calculation, the coefficient of determination for validation ($R^2_{cal}$ and $R^2_{val}$), the root mean square error of calibration (RMSEC), and the root mean square error of validation (RMSEV) were calculated (Filgueiras et al., 2014; Sujaritha et al., 2019), which can select the optimized 15[th] component to obtain the optimal fitting results. In the case of the PLSR

model for $15^{th}$ components in our study, the mean squared error of the prediction (MSEP) of 1.96 $Mm^{-1}$ is low, and the coefficient of determination ($R^2$) of 0.999 is high, representing the best-suited number of components for β predictions (Wehrens and Mevik, 2007; Kvalheim et al., 2019). In addition, as depicted in Figure S4, the predicted value of $b_{abs}$ (i.e., the sum of $Mb_{abs}$ for all measured molecules) was highly correlated with the measured methanol-soluble BrC total $b_{abs}$ (slope = 0.51, $r^2$ = 0.91, p < 0.001). It is noteworthy that over 70% of predicted $b_{abs}$ points displayed more appropriate model performances with increasing measured $b_{abs}$ at full spectrum. Further, the non-zero intercept in Figure S4 was used to indicate a sufficient contribution of other undetected BrC components (i.e., nonpolar or less polar compounds such as saturated hydrocarbons and PAHs), rather than $ESI^-$ selected BrC to the measured absorption (Lin et al., 2018).

③ BrC compounds selection. The identified CHO, CHON, and CHONS compounds discussed in our study can be divided into two categories. One refers to the whole compounds detected using UHPLC-Q-Tof MS/MS, which is only used in the original section 2.3 to obtain their individual $Mb_{abs}$ by fitting the $b_{abs365}$ of MSOC. Certainly, not all detected molecules are BrC that have light absorption at 365 nm. The value of $Mb_{abs}$ was used as a criterion to distinguish whether these measured molecules are BrC compounds or not. In this study, the molecules with high $Mb_{abs}$ ($\geq 10^{-8}$) represent BrC while relatively low $Mb_{abs}$ values ($<10^{-8}$) as non-BrC molecules. Non-BrCs are molecules with negative or extremely low absorptions, which were not discussed in detail in this study because of their negligible absorption contributions. The other category represents the measured CHO, CHON, and CHONS compounds with high $Mb_{abs}$ ($\geq 10^{-8}$), which was identified as BrC molecules and was thoroughly discussed in section 3.2 and 3.3. To avoid misunderstanding, section 2.3 in the original manuscript was divided into two parts of "Section 2.3 Detection of MSOC molecules" and "Section 2.4 Calculation of absorption from individual BrC molecules" in the revised manuscript. In response to the fifth major comment raised by the first reviewer, the related sentences about the classification of discussed BrC molecules were added

to lines 149-151 in the revised manuscript to clarify the identified CHO, CHON, and CHONS compounds.

Furthermore, we totally agreed that the limitations of the molecules' detection exist using negative ESI mode. The polar compounds such as nitro-phenols, aromatic acids, and carboxylic acids can be preferentially ionized by negative ESI mode, while the detection of oxygenated aliphatic and non-polar aromatic compounds (i.e. larger PAHs) can be achieved from positive ESI or atmospheric pressure photoionization (APPI) mode (Cha et al., 2018; Lin et al., 2018). Therefore, as depicted below, these limitations were added to the description of MSOC molecules detection and the interpretation of the slope and intercept for PLSR, respectively.

*Line 114-116:*

*"In this study, the negative ESI UHPLC-Q-TOF MS/MS can measure semi-polar and polar organic molecules with acidic protons, but it is insensitive to non-polar compounds. Therefore, the detected molecules in our study only refer to a part of the MSOC fraction."*

*Line 153-156:*

*"The slope here suggests that the model explained nearly 60% of the measured $b_{abs}$ in the full spectrum, and the nonzero intercept in linear correlations indicates a contribution of undetected non-polar or weakly polar organic compounds using a negative ESI source."*

References:

Cha, E., Jeong, E. S., Han, S. B., Cha, S., Son, J., Kim, S., Oh, H. B., and Lee, J.: Ionization of gas-phase polycyclic aromatic hydrocarbons in electrospray ionization coupled with gas chromatography, Anal. Chem., 90, 4203-4211, 10.1021/acs.analchem.8b00401, 2018.

Filgueiras PR, Alves JCL, Sad CM, Castro EV, Dias JC, Poppi RJ.: Evaluation of trends in residuals of multivariate calibration models by permutation test. Chemometr. Intell. Lab, 133:33-41, 10.1016/j.chemolab.2014.02.002, 2014.

Kvalheim, O.M., Grung, B., Rajalahti, T.: Number of components and prediction error in partial least

*squares regression determined by Monte Carlo resampling strategies. Chemometr. Intell. Lab, 188, 79–86, 10.1016/j.chemolab.2019.03.006, 2019.*

*Lin, P., Fleming, L.T., Nizkorodov, S.A., Laskin, J., Laskin, A.: Comprehensive molecular characterization of atmospheric brown carbon by high resolution mass spectrometry with electrospray and atmospheric pressure photoionization. Anal. Chem., 90, 12493-12502, 10.1021/acs.analchem.8b02177, 2018.*

*Man, Y., Stenrød, M., Wu, C., Almvik, M., Holten, R., Clarke, J. L., Yuan, S., Wu, X., Xu, J., Dong, F., Zheng, Y., and Liu, X.: Degradation of difenoconazole in water and soil: Kinetics, degradation pathways, transformation products identification and ecotoxicity assessment. J. Hazard. Mater., 418, 126303, 10.1016/j.jhazmat.2021.126303, 2021.*

*Sujaritha, M., Kavitha, K., Janet, J.: Comparative analysis of accuracy on partial least squares and principal component regression methods. Indian J. Sci. Technol., 12, 0974-6846, 10.17485/ijst/2019/v12i8/141809, 2019.*

*Wehrens, R., Mevik, B.H.: The pls package: Principal component and partial least squares regression in r. J. Stat. Softw., 18, 1-24, hdl.handle.net/2066/36604, 2007.*

*Xu, L., Liu, Y., Wu, H., Wu, H., Liu, X., and Zhou, A.: Rapid identification of chemical profile in Gandou decoction by UPLC-Q-TOF-MSE coupled with novel informatics UNIFI platform. J. Pharm. Anal., 10(1), 35-48, 10.1016/j.jpha.2019.05.003, 2020.*

7. The manuscript should be proofread carefully. For example, page 2, line 58, "liner correlations".

**Response**: Suggestion taken. We have corrected "liner correlations" as "linear correlations".